# CausalDiff: Causality-Inspired Disentanglement via Diffusion Model for Adversarial Defense

**Mingkun Zhang**
CAS Key Laboratory of AI Safety
Institute of Computing Technology, CAS
zhangmingkun20z@ict.ac.cn

**Keping Bi**
Key Laboratory of Network
Data Science and Technology
Institute of Computing Technology, CAS
bikeping@ict.ac.cn

**Wei Chen** *
CAS Key Laboratory of AI Safety
Institute of Computing Technology, CAS
chenwei2022@ict.ac.cn

**Quanrun Chen**
School of Statistics University
of International Business and Economics
qchen@uibe.edu.cn

**Jiafeng Guo**
Key Laboratory of Network
Data Science and Technology
Institute of Computing Technology, CAS
guojiafeng@ict.ac.cn

**Xueqi Cheng**
CAS Key Laboratory of AI Safety
Institute of Computing Technology, CAS
cxq@ict.ac.cn

## Abstract

Despite ongoing efforts to defend neural classifiers from adversarial attacks, they remain vulnerable, especially to unseen attacks. In contrast, humans are difficult to be cheated by subtle manipulations, since we make judgments only based on essential factors. Inspired by this observation, we attempt to model label generation with essential label-causative factors and incorporate label-non-causative factors to assist data generation. For an adversarial example, we aim to discriminate the perturbations as non-causative factors and make predictions only based on the label-causative factors. Concretely, we propose a casual diffusion model (CausalDiff) that adapts diffusion models for conditional data generation and disentangles the two types of casual factors by learning towards a novel casual information bottleneck objective. Empirically, CausalDiff has significantly outperformed state-of-the-art defense methods on various unseen attacks, achieving an average robustness of 86.39% (+4.01%) on CIFAR-10, 56.25% (+3.13%) on CIFAR-100, and 82.62% (+4.93%) on GTSRB (German Traffic Sign Recognition Benchmark). The code is available at https://github.com/CAS-AISafetyBasicResearchGroup/CausalDiff.

## 1 Introduction

Neural classifiers, despite their impressive performance in various applications, are susceptible to adversarial attacks [1, 2], which can deceive them into making erroneous judgments on subtly manipulated examples. Such vulnerabilities pose severe threats in safety-critical scenarios such as face recognition [3, 4] and autonomous driving [5]. There has been extensive work on defending against adversarial attacks such as certified defenses [6, 7], adversarial training [8, 9], and adversarial purification Samangouei et al. [10].

---

*Corresponding Author.
CAS stands for Chinese Academy of Sciences

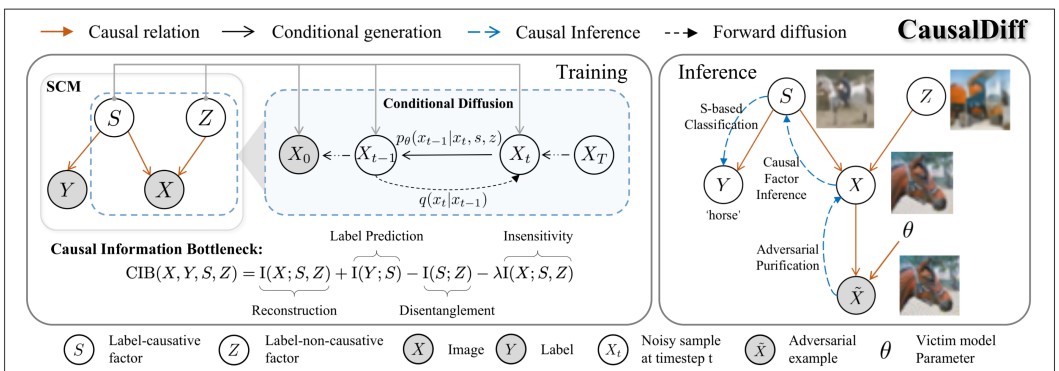

Figure 1: Illustration of training (Left) and inference (Right) processes of our proposed CausalDiff model. During training, the model constructs a structural causal model leveraging a conditional diffusion model, disentangling the (label) Y-causative feature $S$ and the Y-non-causative feature $Z$ through maximization of the Causal Information Bottleneck (CIB). In the inference stage, CausalDiff first purifies an adversarial example $\tilde{X}$, yielded by perturbing $X$ according to the target victim model parameterized by $\theta$, to obtain the benign counterpart $X^*$. Then, it infers the Y-causative feature $S^*$ for label prediction. We visualize the vectors of $S$ and $Z$ inferred from a perturbed image of a horse using a diffusion model. We find that $S$ captures the general concept of a horse, even when the input image only shows the head, while $Z$ carries information about the horse's skin color.

Although effective, these methods have some limitations. Certified defense methods have limited practicality due to the small certified region that can theoretically guarantee robustness. Adversarial training approaches suffer from a significant decline in robustness against unseen attacks since they take effect by adding adversarial examples into the training set. Purification methods, not designed for specific attacks, struggle to determine the optimal denoising level for unforeseen attacks with differing degrees of perturbation. Consequently, they face challenges in effectively defending against unseen attacks.

In reality, attack behaviors are often unpredictable. Is there a way of strengthening a model to act like humans, i.e., be insensitive to subtle perturbations and robust against various unforeseen attacks? Given an image of an object, we typically identify the key visual features that are necessary to determine its category and disregard other factors such as styles, backgrounds, details, or perturbations. This allows us to make robust judgments. Inspired by this human decision-making process, we would like to learn a model that can disentangle the essential features for determining the category from other non-essential ones.

It is natural to treat the essential features as the causal factors of the label, and both essential and the other features as the causal factors of the entire image. Then, we can learn to disentangle them by modeling the process of data generation and label prediction with a structural causal model (SCM) [11, 12](shown in Fig. 1 (Left)). According to the theoretical results provided by Liu et al. [13], the identifiability of such SCMs can be guaranteed under mild conditions. Although similar SCMs have also been employed in modeling the generation of multi-domain data [14] and adversarial examples [15, 16], they either aim to enhance out-of-distribution robustness or protect the model from a certain type of attack. In contrast, our research focuses on modeling the generation of native in-domain data to enhance adversarial robustness against various unseen attacks.

Fig. 1 (Left) depicts our SCM, where $Y$ denotes the category (e.g., horse) of an input image $X$; $S$ denotes the essential semantic features of determining $Y$ (i.e., $Y$-causative factors), such as the characteristics of eyes, ears, nose, mouth, etc. of horses; and $Z$ represents the other features (i.e., $Y$-non-causative factors) that are not needed to predict $Y$ but are important to generate $X$, such as the fur color and the image background. Given an adversarial example produced by an unknown attack, our model aims to disentangle its $Y$-causative features $S$ from the spurious factors in $Z$ and make a robust prediction. To successfully learn the disentanglement, our SCM is guided by the tasks of data generation and label prediction, where there are three major challenges:

1) How do we model the conditional generation of $X$ given $S$ and $Z$ effectively? To this end, we employ a well-recognized diffusion model with state-of-the-art (SOTA) generative performance and

efficiency, i.e., the Denoising Diffusion Probabilistic Model (DDPM) [17], as the backbone. We further adapt it for conditional generation from latent variables rather than random noise. 2) What training objective should we use to effectively learn the disentanglement of $S$ and $Z$? With this regard, we propose a Causal Information Bottleneck (CIB) optimization objective. CIB aligns the information in the latent variables $(S, Z)$ with observed variables $(X, Y)$ with a bottleneck set by the mutual information (MI) between $S, Z$ and $X$. The derived function will minimize the MI between $S$ and $Z$ while learning the other causal relations, ensuring their disentanglement within the causal framework. 3) Given an adversarial example, what inference strategies shall we adopt to make robust predictions? As shown in Fig. 1 (Right), according to how the adversarial example $\tilde{X}$ is generated, we first purify it to yield a benign example $X^*$ and then infer the Y-causative factor $S$ based on $X^*$ for final classification. We name the entire causal defense framework based on diffusion models as CausalDiff.

The experimental results on facing various unseen attacks, encompassing both black-box and white-box ones, show that CausalDiff has superior performance compared to representative adversarial

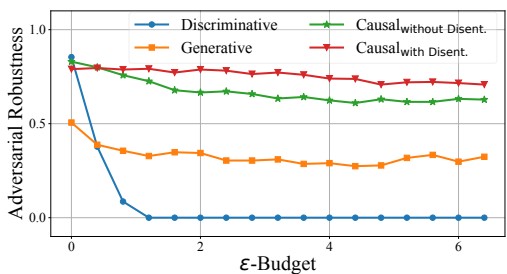

Figure 2: Adversarial robustness of four models against 100-step PGD attack under varying attack strength indicated by $\epsilon$-budget.

Table 1: The experimental results of four models on toy data. The variation of latent $v$ and logits $p(y|v)$ is measured between clean and adversarial examples. The model margin is estimated by the minimal adversarial perturbation required to flip the label, employing both $\ell_2$ and $\ell_\infty$ norm.

| Model | $\triangle v \downarrow$ | $\triangle p(y\|v) \downarrow$ | margin $\uparrow$ ($\ell_2$ / $\ell_\infty$) |
|---|---|---|---|
| Discriminative | 1.15 | 0.81 | 2.14 / 0.38 |
| Generative | **0.06** | 0.27 | 1.12 / 0.24 |
| Causal$_{\text{w/o Disent.}}$ | 0.29 | 0.32 | 9.58 / 5.30 |
| Causal$_{\text{w/ Disent.}}$ | 0.27 | **0.22** | **10.64 / 6.28** |

defense baselines including state-of-the-art (SOTA) methods. Specifically, our CausalDiff achieves robustness of 86.39% (+4.01%) on CIFAR10, 56.25% (+3.13%) on CIFAR-100 [18], and 81.79% (+4.93%) on GTSRB [19].

In summary, we highlight our contributions as follows: 1) We propose a novel causal diffusion framework (CausalDiff) to defend against unseen attacks by modeling the generation of native in-domain data with the category(Y)-causative factors and the other Y-non-causal factors; 2) We propose a Causal Information Bottleneck (CIB) objective to disentangle Y-causative from Y-non-causative factors during causal model training and an associated inference algorithm for adversarial defense; 3) CausalDiff significantly outperforms SOTA methods in defending against various unseen attacks.

## 2 Related Work

**Adversarial Defense.** Adversarial training primarily focuses on optimizing the training algorithm [8, 20, 21], incorporating data augmentation [22, 23, 9], and enhancing acceleration [24, 25]. Despite its effectiveness, the trained models could still be vulnerable to unseen attacks [26, 27]. Adversarial purification, orthogonal to our work, utilizes a generative model to purify adversarial noise from examples before classification. Leveraging diffusion models [28, 17, 29, 30], diffusion-based purification have shown to be effective [31–35, 27].

**Causal Learning for Robustness.** Causal representation learning [12, 36, 14, 37] focuses on discovering invariant mechanisms within structural causal models and has achieved remarkable performance in improving model transferability [14, 13, 38–40] and interpretability [41, 42]. Moreover, in terms of adversarial robustness, researchers [43, 15, 44, 45, 16, 46–48] have attempted to model the attack behaviors in causal structures to identify the adversarial factors. However, modeling particular attack types will limit the model robustness on other types of attacks [27].

**Conditional Diffusion Model.** Diffusion models [28, 17, 29, 30, 49] have achieved compelling image generation capabilities. To equip them with the ability of controllable generation, the sampling

process can be guided by 1) classifier confidence to generate images of a certain category [30, 50, 51], and 2) semantic embedding of text content or a certain style [52–55].

## 3 Pilot Study on Toy Data

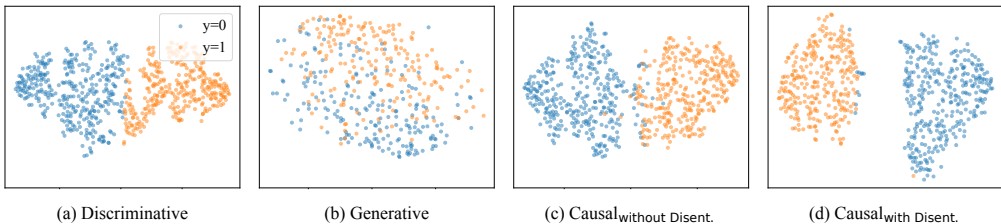

(a) Discriminative        (b) Generative        (c) Causal$_{without\ Disent.}$        (d) Causal$_{with\ Disent.}$

Figure 3: Visualizations of feature space for the two categories on toy data by T-SNE for (a) discriminative model, (b) generative model, (c) causal model without disentanglement, and (d) causal model with disentanglement.

To compare our proposed causal model with other representative models in terms of adversarial robustness, we constructed a toy dataset according to the hypothesis that the essential factors are the basis of generating labels and they also determine the generation of data together with label-non-causative features. Pilot studies on this toy data will provide insights into whether our causal model will work and why it works.

### 3.1 Experimental Settings

**Toy Data Construction.** We constructed 2000 samples following the causal structure in Fig. 1 (Left). Specifically, for each data point, we randomly sample the vectors $s$ and $z$ from two different normal distributions, project $s$ to a score $y_s$ with random weights, obtain its label by comparing $y_s$ with the medium score, and generate the representation $x$ by projecting the concatenation $[s; z]$ with another random matrix. Please refer to Appendix B.3 for detailed information.

**Models for Comparisons.** We investigated four representative models: 1) a *Discriminative* model for classification, 2) a *Generative* model that predicts the label of an adversarial example with $\max_y p(x|y)$ [27], 3) a causal model without disentanglement (*Causal$_{without\ Disent.}$*) that models the generation of both x and y with the same latent factor $v$, 4) our causal model with disentanglement (*Causal$_{with\ Disent.}$*) that is illustrated in Fig. 1. The concrete structures of the four models and further details are presented in Appendix B.3.

### 3.2 Experimental Observations

**Adversarial Robustness.** We evaluate the model's robustness against a 100-step PGD attack with varying $\epsilon$-budgets under the $\ell_\infty$ bound [8]. Model performance in terms of both clean accuracy (when $\epsilon = 0$) and robust accuracy is examined.

As shown in Fig. 2: 1) The *Discriminative* model exhibits the highest clean accuracy but suffers a rapid decline in robustness, dropping to $0\%$ when the attack budget $\epsilon$ reaches 1.2., which highlights its vulnerability. 2) The *Generative* model has the lowest clean accuracy while its robustness does not dramatically regress with larger attack budgets. The low clean accuracy may be due to the inconsistency between the generation process it modeled and the way this toy dataset is constructed. The small gap between clean and robust accuracy indicates its decent effectiveness in defending against adversarial attacks. 3) *Causal$_{with\ Disent.}$* obtains the second-best clean accuracy, yet its robustness gradually declines with increasing attack strength, maintaining 61.4% robustness at $\epsilon = 6.4$. 4) Our model, *Causal$_{with\ Disent.}$*, has slightly lower clean accuracy than Causal without Disent. but the best robustness among the four. As the attack strength increases, it maintains at least 71.8% robust accuracy at $\epsilon = 6.4$, indicating that it is promising to enhance model robustness by causal modeling with disentanglement of the label-causative factors from the non-causative ones.

**Sensitivity to Perturbations.** To delve further into how the model behavior changes when defending against adversarial perturbations, we measure the variation between latent variables of clean and adversarial examples, denoted as $\triangle v = 1 - cosine(v, v_{\text{adv}})$, where $v$ and $v_{\text{adv}}$ represents the latent vector of clean and adversarial example, respectively ($v = s$ for $\text{Causal}_{with\ Disent.}$). We also compute the change of predicted logits, $\triangle p(y|v) = p(y|v) - p(y|v_{\text{adv}})$, with $y$ being the true class label. Note that a larger variation in the latent factor for prediction, or the predicted logits, results in increased insensitivity to perturbations.

The experimental results, as shown in Tab. 4, indicate that $\text{Causal}_{with\ Disent.}$, compared to $\text{Causal}_{without\ Disent.}$, exhibits less sensitivity in both latent variables and prediction outcomes. This suggests that by disentangling the label-causative factor $s$, it becomes more challenging for attackers to perturb the model and alter its prediction. The small variation of $v$ in the Generative model is probably attributed to the lack of discriminability in $v$.

**Prediction Margins.** To intuitively explain the sensitivity of each model to perturbations, we estimate the minimal adversarial perturbation $\|\delta\|$ under PGD required to flip the correct model prediction $y$ of a sample $x$. It can be interpreted as the prediction margins in terms of perturbation, denoted as $\text{margin}(x, y) = \min \|\delta\|$, subject to $p(y|x + \delta) < p(\bar{y}|x + \delta)$ [56]. We measure the $\|\delta\|$ under $\ell_2$ and $\ell_\infty$ norms. Additionally, we visualized the latent vector space of each model to intuitively observe the margin of the classification boundary.

As shown in Tab. 4, $\text{Causal}_{with\ Disent.}$ has the largest prediction margin. This implies that an attacker would need to add significantly more perturbation to successfully cheat our model. We can draw similar conclusions from Fig. 3, which illustrates the distribution of predictive features extracted by each model for correctly classified clean samples across categories. Again, $\text{Causal}_{with\ Disent.}$ has the largest margin between the classes, indicating that it has high confidence in its prediction and increases the cost and difficulty for an attacker to succeed.

## 4 Causal Diffusion Model

To enhance model robustness on real-world data, in this section, we propose a *Causal Diffusion (CausalDiff)* Modeling approach, that couples our previously studied SCM (shown in Fig. 6(d)) with diffusion models. We will introduce the three major components in CausalDiff: conditional diffusion generation, causal information bottleneck optimization, and adversarial example inference. We take the Denoising Diffusion Probabilistic Model (DDPM) [17] as an instance for illustration and it can be easily adapted to other diffusion models.

### 4.1 Conditional Diffusion Generation

Standard diffusion models generate images based on random noise which do not apply to the conditional generation of $X$ based on $S$ and $Z$ in our SCM. In standard diffusion models, at each time step $t$ during denoising, a UNet $\epsilon_\theta(x_t, t)$ is employed to decode an image given input $x_t$. While diffusion models are highly effective in image generation, they lack an explicit decoder component for generating images from latent variables. To handle this, we develop a conditional DDPM using latent variables $S$ and $Z$ controlling the generation process. This approach draws inspiration from both the class-conditional diffusion model [51] as well as the style control mechanism introduced in DiffAE [52]. The output $h_{\text{out}}^t$ at each layer of the UNet depends on $t$ and $x_t$, i.e.,

$$h_{\text{out}}^t = t_{\text{s}} \cdot \text{GroupNorm}(h) + t_{\text{b}}, \tag{1}$$

where $h$ is the feature map of $x_t$, $t_{\text{s}}$ and $t_{\text{b}}$ are the scale and bias of timestep $t$.

Inspired by the class-conditional diffusion model [51] and the style control mechanism in DiffAE [52], we adapt the standard diffusion generation to be conditioned on $S$ and $Z$ in addition to timestep $t$. Specifically, the UNet becomes $\epsilon_\theta(x_t, t, s, z)$, the final output $h_{\text{out}}^{t,s,z}$ is calculated based on the original $h_{\text{out}}^t$, the hidden state $s$ and $z$ (representing causal factor $S$ and $Z$) encoded from input $x$:

$$h_{\text{out}}^{t,s,z} = z_{\text{s}} \cdot h_{\text{out}}^t + s_{\text{b}}, \tag{2}$$

where $z_s$ and $s_b$ are produced from the affine projections of $z$ and $s$ respectively, i.e., $z_{\text{s}} = \text{Affine}_z(z)$ and $s_{\text{b}} = \text{Affine}_s(s)$. As such, the label-causative factor $S$ acts as a bias that can affect the direction of the latent vector and change its semantics, while the label-non-causative factor $Z$ can only scale the latent vector in a similar way to style control.

## 4.2 Causal Information Bottleneck Optimization

To learn the causal factors $S, Z$ in our SCM and disentangle them, we propose a Causal Information Bottleneck (CIB) optimization objective. It maximizes the mutual information between the latent factors $S, Z$ and the observed data sample $(X, Y)$ with an information bottleneck that constrains the information retained in $S, Z$ with respect to $X$.

Specifically, to align the information captured in the latent factors with the observed data $(X, Y)$, we maximize the mutual information between them, denoted as $\mathrm{I}(X, Y; S, Z)$, which can be derived as:

$$\mathrm{I}(X, Y; S, Z) = \mathrm{I}(X; S, Z) + \mathrm{I}(Y; S) - \mathrm{I}(S; Z) - \mathrm{I}(X; Y). \tag{3}$$

A detailed proof is presented in Appendix A.1. Among the resultant terms, $\mathrm{I}(X; Y)$ is solely dependent on the observed data, independent of latent variables or the causal model, and thus can be ignored in the learning process. Maximizing $\mathrm{I}(X; S, Z)$ will urge $S$ and $Z$ to capture ample information about $X$. $\mathrm{I}(S; Y)$ indicates that the $Y$-causative factor $S$ should be correlated with $Y$. The term, $-\mathrm{I}(S; Z)$ ensures $S$ and $Z$ to be effectively disentangled.

Existing work on optimizing similar SCMs adapts the Evidence Lower BOund (ELBO) from Variational Autoencoders (VAE) to formulate the causal ELBO objectives [14]. This objective only maximizes the likelihood of $(X, Y)$, and does not consider the latent factors. Consequently, the final optimization goal of causal ELBO differs from $I(X, Y; S, Z)$ in that it does not have $-\mathrm{I}(S; Z)$ in Eq. (3), which is crucial for disentanglement. Further details are discussed in Appendix A.4.

To avoid $X$ contain too many unimportant details, we constrain the mutual information between $X$ and the latent factors $S, Z$ with an information bottleneck $\mathrm{I}_c$. Then, the updated objective becomes:

$$\max \mathrm{I}(X, Y; S, Z), \quad s.t. \mathrm{I}(X; S, Z) \leq \mathrm{I}_c. \tag{4}$$

Employing Lagrange multiplier $\lambda \geq 0$, we formulate our objective as $\max \mathrm{I}(X, Y; S, Z) - \lambda(\mathrm{I}(X; S, Z) - \mathrm{I}_c)$. Since $I_c$ is a constant, it is equal to maximize the **Causal Information Bottleneck** (CIB):

$$\mathrm{CIB}(X, Y, S, Z) = \mathrm{I}(X; S, Z) + \mathrm{I}(Y; S) - \mathrm{I}(S; Z) - \lambda \mathrm{I}(X; S, Z). \tag{5}$$

$I(X; S, Z)$ and $-\lambda I(X; S, Z)$ indicate two opposing optimization directions. Because it is unclear whether $(1 - \lambda)$ should be positive or negative, we approximate these terms via two separate lower bounds instead of combining them.

To maximize the Causal Information Bottleneck (CIB) in Eq. (5), we derive its lower bound as the concrete training loss function. When using the diffusion model $\epsilon_\theta$, classifier $f_y(s; \theta)$ (to estimate $p_\theta(y|s)$), and the encoder $f_{s,z}(x; \theta)$ (to estimate $p_\theta(s, z|x)$ ), the lower bound of CIB is:

$$\mathbb{E}_{p(x,s,z)}[\log p_\theta(x|s, z)] + \mathbb{E}_{p(y,s)}[\log p_\theta(y|s)] - \mathrm{I}^\theta_{\mathrm{CLUB}}(S; Z) - \lambda \mathbb{E}_{p(x)}[\mathcal{D}_{\mathrm{KL}}(p_\theta(s, z|x) \| q(s, z))], \tag{6}$$

where $p_\theta(s|z)$ is a variational distribution to estimate $p(s|z)$ and $\mathrm{I}^\theta_{\mathrm{CLUB}}(S; Z) = \mathbb{E}_{p(s,z)}[\log p_\theta(s|z)] - \mathbb{E}_{p(z)}\mathbb{E}_{p(s)}[\log p_\theta(s|z)]$ represents the Contrastive Log-Ratio Upper Bound (CLUB) of mutual information proposed by Cheng et al. [57]. $p_\theta(x|s, z)$ represents the likelihood estimated by the conditional diffusion model. $\mathcal{D}_{\mathrm{KL}}$ refers to the Kullback-Leibler (KL) divergence [58]; and $q(\cdot)$ denotes the prior distribution of the latent variable, e.g., a normal distribution $\mathcal{N}(\mathbf{0}, \mathbf{I})$. A detailed proof can be found in Appendix A.2.

**Loss Function**. Thus, maximizing the lower bound of CIB is equal to minimizing the loss function:

$$\begin{aligned} \mathcal{L}(x, y, s, z; \theta) = {} & \alpha \mathbb{E}_{\epsilon,t} \| \epsilon_\theta(x_t, t, s, z) - \epsilon_t \|_2^2 + \gamma \mathcal{L}_{\mathrm{CE}}(s, y; \theta) \\ & + \eta \mathrm{I}^\theta_{\mathrm{CLUB}}(S; Z) + \lambda \mathcal{D}_{\mathrm{KL}}(p_\theta(s, z|x) \| q(s, z)), \end{aligned} \tag{7}$$

where $\alpha, \lambda, \gamma, \eta$ determine the weighting of each term in the optimization process. A detailed derivation can be found in Appendix A.3.

**Algorithm**. We pretrain the model with the data reconstruction loss (the first term in Eq. (7)) alone before training the model with the entire loss, so that the model can learn the causal factors, disentanglement, and classification from a decent starting point. For space concern, we illustrate the training process in Appendix B.2 involves the training algorithm (Algorithm 1) and the pretraining algorithm (Algorithm 2).

Table 2: Clean accuracy and adversarial robustness on CIFAR-10 against **StAdv** under $\ell_\infty$ ($\epsilon = 0.05$) norm bound and **AutoAttack (AA)** under $\ell_2$ ($\epsilon = 0.5$) as well as $\ell_\infty$ ($\epsilon = 8/255$) bound. We calculate the average robustness across three attack methods to evaluate the model's robustness against unseen attacks. We use underlining to highlight the best robustness for each attack method within each defense category, and bold font to denote the state-of-the-art (SOTA) across all methods.

| | METHOD | BACKBONE | CLEAN ACC (%) | AA $l_\infty$ | AA $\ell_2$ | STADV | AVG |
|---|---|---|---|---|---|---|---|
| | | | | | ROBUST ACC(%) | | |
| ADV. TRAIN | AT-$l_\infty$ [23] | DDPM | 88.87 | 63.28 | 64.65 | 4.88 | 44.27 |
| | AT-$l_2$ [23] | DDPM | 93.16 | 49.41 | 81.05 | 5.27 | 45.24 |
| | AT-$l_\infty$ [9] | EDM | 93.36 | 70.90 | 69.73 | 2.93 | 47.85 |
| | AT-$l_2$ [9] | EDM | 95.90 | 53.32 | 84.77 | 5.08 | 47.72 |
| | CAUSALADV-T [15] | WRN-76-10 | 83.71 | 8.76 | 21.95 | 75.60 | 35.44 |
| | CAUSALADV-M [15] | WRN-76-10 | 70.22 | 24.36 | 49.10 | 48.60 | 40.69 |
| | DICE [16] | WRN-34-10 | 82.85 | 37.51 | 41.58 | 82.46 | 53.85 |
| PURIFY | DIFFPURE [33] | SCORE SDE | 87.50 | 53.12 | 75.59 | 12.89 | 47.20 |
| | LM-DDPM[27] | DDPM | 80.47 | 53.32 | 63.09 | 74.22 | 63.54 |
| | LM-EDM[27] | EDM | 87.89 | 71.68 | 75.00 | 87.50 | 78.06 |
| OTHERS | SBGC [59] | SCORE SDE | 95.04 | 0.00 | 0.00 | 0.00 | 0.00 |
| | CAMA [43] | VAE | 32.19 | 3.38 | 5.53 | 27.54 | 12.15 |
| | RDC [27] | EDM | 89.85 | 75.67 | 82.03 | 89.45 | 82.38 |
| OURS | **CAUSALDIFF** | DDPM | 90.23 | **83.01** | **86.33** | 89.84 | **86.39** |
| | **CAUSALDIFF** w/o CFI | DDPM | 83.20 | 74.61 | 75.59 | 82.23 | 77.48 |
| | **CAUSALDIFF** w/o AP | DDPM | 91.21 | 69.14 | 84.96 | **91.21** | 81.77 |

## 4.3 Adversarial Example Inference

Guided by the causal generation of an adversarial example $\tilde{X}$ according to Fig. 1 (Right), we illustrate the process for robust classification. Following a typical attack paradigm, $\tilde{X}$ is produced by adding an adversarial perturbation to a target clean example $X$ when attacking a model $\theta$. To make a robust prediction on $\tilde{X}$, our robust inference process comprises three steps: 1) purifying $\tilde{X}$ to benign $X$ by the unconditional diffusion model $\epsilon_\theta(x_t, t)$, 2) inferring $S$ and $Z$ from $X$ utilizing the causal model $\epsilon_\theta(x_t, t, s, z)$, and 3) predicting $Y$ based on $S$ using a classifier $f_y(s; \theta)$.

**Adversarial Purification (AP).** We follow the concept of Likelihood Maximization (LM) [33, 27] to purify the adversarial example $\tilde{X}$ to a benign $X^*$ by maximizing the data log-likelihood $\log p_\theta(x)$:

$$x^* = \arg\max_x \log p_\theta(\tilde{x}). \tag{8}$$

Concretely, we maximize its lower bound utilizing the unconditional diffusion model $\epsilon_\theta(x_t, t)$ trained according to Section C.2. Chen et al. [27] suggest using one random timestep $t$ during each purification iteration while we believe that smaller timesteps should be more effective since they retain more information from the original example. Thus, we limit the random selection to within the first 50 timesteps. This way significantly boosts adversarial robustness, which will be discussed in Section 5.3.

**Causal Factor Inference (CFI).** In order to infer the causal and non-causal factors $S$ and $Z$, which can reconstruct the original image $X$, we optimize the latent variables by maximizing the conditional likelihood $p_\theta(x|s, z)$ employing the trained conditional diffusion model $\epsilon_\theta(x_t, t, s, z)$:

$$s^*, z^* = \arg\max_{s,z} \log p_\theta(x^*|s, z). \tag{9}$$

Similarly to purification, we obtain $s^*$ and $z^*$ by maximizing the lower bound $-\mathbb{E}_{\epsilon,t}[w_t\|\epsilon_\theta(x_t, t, s, z) - \epsilon\|]$ using the conditional diffusion model $\epsilon_\theta(x_t, t, s, z)$. For efficiency concerns, instead of using all the timesteps for estimation as in Chen et al. [27], we sample $N_{\text{purify}} = 5$ timesteps at the same intervals across the entire timesteps.

**Latent-S-Based Classification (LSBC).** After obtaining $s^*$ according to Eq. (9), we use the trained classifier $f_y(s; \theta)$ to predict label $Y$:

$$y^* = \arg\max_y \log p_\theta(y|s^*). \tag{10}$$

Table 3: Clean accuracy and adversarial robustness against **AutoAttack (AA)** on **GTSRB (Left)** and **CIFAR-100 (Right)** dataset. We use $\epsilon = 8/255$ as $\ell_\infty$ and $\epsilon = 0.5$ as $\ell_2$ norm bound.

| Method | Clean Acc | Robust Acc | | | |
|---|---|---|---|---|---|
| | | AA $l_\infty$ | AA $\ell_2$ | Fog | Avg |
| DOA [61] | 76.56 | 31.25 | 36.72 | 68.36 | 45.44 |
| GTSRB-CNN [3] | 93.95 | 62.30 | 74.80 | 65.43 | 67.51 |
| AT-4 [8] | 92.58 | 74.78 | 80.47 | 78.13 | 77.69 |
| AT-8 [8] | 91.21 | 74.02 | 79.10 | 73.44 | 75.52 |
| AT-16 [8] | 89.65 | 73.24 | 75.59 | 69.92 | 72.92 |
| **CausalDiff** | 97.85 | **80.86** | **80.86** | **86.13** | **82.62** |

| Method | Clean Acc | Robust Acc |
|---|---|---|
| WRN40-2 | 78.13 | 0.00 |
| AT-DDPM [23] | 63.56 | 34.64 |
| AT-EDM [9] | 75.22 | 42.67 |
| DiffPure [33] | 39.06 | 7.81 |
| DC [27] | 79.69 | 39.06 |
| RDC [27] | 80.47 | 53.12 |
| **CausalDiff** | 65.62 | **56.25** |

Within our inference pipeline, both adversarial purification and causal factor inference leverage the diffusion model learned toward the CIB optimization objective while they take effect independently. When we combine these two approaches, adversarial robustness could be enhanced further. The concrete inference algorithm in Algorithm 3 is presented as Appendix B.2.

### 4.4 Comparison with Adversarial Purification

First, **our CausalDiff can be viewed as semantic-level purification**. Instead of pixel-level denoising, CausalDiff purifies an image in the latent space, trying to remove the effect of perturbation by putting it to the label-non-causative features. Second, conventional purification inevitably loses information essential for classification during denoising. By disentangling label-causative features from label-non-causative features, CausalDiff can retain essential information in the label-causative features to a large extent. Third, unlike pixel-level purification which does not know the optimal denoising level for various attacks, CausalDiff acts adaptively on different attacks by the causal inference of $S$ and $Z$. Fourth, they can be combined to further enhance adversarial robustness.

## 5 Experiments

In this section, we introduce the experimental settings in Section 5.1. Section 5.2 presents the main results defending against unforeseen attacks on CIFAR-10, CIFAR-100, and GTSRB datasets. We then evaluate the effectiveness of individual components of CausalDiff in Section 5.3. Due to space constraints, we provide analyses of core components during training and inference in Appendix C.1 and Appendix C.2. Additionally, we showcase examples in Appendix C.3.

### 5.1 Experimental Settings

**Datasets and Model Architecture.** Our experiments utilize the CIFAR-10, CIFAR-100 [18] and GTSRB [19] datasets. CIFAR-10 and CIFAR-100 each consists of 50,000 training images, categorized into 10 and 100 classes, respectively. GTSRB comprises 39,209 training images (each histogram equalized and resized to $3 \times 32 \times 32$) of German traffic signs, categorized into 43 classes. We use DDPM [17] as the diffusion model. Further details are available in Appendix B.2.

**Attack Evaluation Method.** For the CIFAR-10 dataset, we utilize **seven types of attack strategies** with both $\ell_\infty$ and $\ell_2$ norm bounds for evaluation. These strategies include StAdv attack [60], BPDA+EOT, and AutoAttack [2] (AA), which comprises white-box attacks such as APGD-ce, APGD-t, FAB-t, and a black-box Square Attack. For CIFAR-100, we follow the setting of Chen et al. [27], evaluating against the $\ell_\infty$ threat model with $\epsilon = 8/255$. For the GTSRB dataset, we utilize **four types of attacks as well as fog corruptions**. Attack methods include AutoAttack, which comprises APGD-ce, APGD-t, FAB-t, and Square Attack.

### 5.2 Comparisons on Unseen Attacks

**CIFAR-10**. From the experimental results for CIFAR-10 presented in Table 2, we have several observations: 1) The adversarial training methods perform robustly on the same type of attacks they use for training but poorly on other unseen attacks. The only exception is that when the models employ

the most effective attack (i.e., AT-$\ell_\infty$) for adversarial training, the robustness regarding $\ell_2$ is not hurt much. 2) In contrast, purification methods, especially the ones based on more powerful diffusion models (DDPM and EDM), have decent robustness on each unseen attack and much better average robustness. This is reasonable since they learn to purify the adversarial samples without targeting any specific attacks. 3) Our CausalDiff performs the best not only regarding the average robustness but also on each type of attack. Remarkably, the average robust accuracy (86.39%) is only 3.84% lower than the clean accuracy (90.23%) and it boosts the robustness on the most challenging attack ($\ell_\infty$) to 83.01%. Notably, the reported CausalDiff is based on DDPM, which acts less effectively than EDM, which can be seen from AT and LM purification. Grounded on a stronger backbone, CausalDiff could achieve even better performance. Table 4 shows the performance under the BPDA + EOT (EOT = 20) attack. Under this attack, models whose gradients are not accessible and also be compared (e.g., ADP [31]). The results again confirm the superior performance of CausalDiff across different attack types.

**CIFAR-100**. The experimental results on CIFAR-100 in Table 3 (Right) indicate that CausalDiff achieves the highest robustness among all, even with a much lower clean accuracy. The low clean accuracy is likely due to the insufficient training samples to learn $S, Z$, and its weaker backbone -

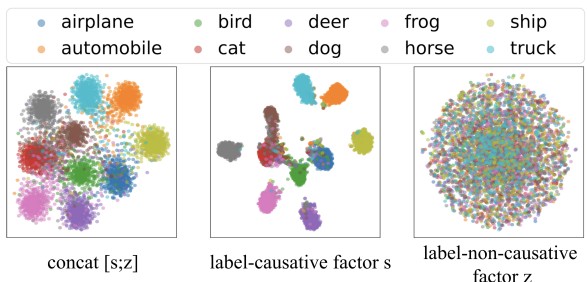

concat [s;z]     label-causative factor s     label-non-causative factor z

Figure 4: Visualization by T-SNE of the feature space, inferred by our CausalDiff, of the label-causative factor $s$, label-non-causative factor $z$, and their concatenation.

Table 4: Clean and robust accuracy on CIFAR-10 against **BPDA + EOT** against $\ell_\infty$ ($\epsilon = 8/255$) threat model.

| Method | Clean Acc (%) | Robust Acc (%) |
|---|---|---|
| Purify - EBM [62] | 84.12 | 54.90 |
| LM - DDPM [27] | 83.20 | 69.73 |
| ADP [31] | 86.14 | 70.01 |
| RDC [27] | 89.85 | 75.67 |
| GDMP [35] | 93.50 | 76.22 |
| DiffPure [33] | 89.02 | 81.40 |
| **CausalDiff** | 90.23 | **88.48** |

DDPM. We believe more augmented data and a stronger diffusion backbone like EDM (used by RDC [27]) could further enhance the performance.

**GTSRB**. The left part of Table 3 show that CausalDiff also has compelling robustness on traffic sign classification, in terms of not only unforeseen adversarial attacks but also natural corruptions like fog. Evaluations based on different types of tasks, and different numbers of classes (10, 100, and 43) have all shown the efficacy of CausalDiff.

## 5.3 Ablation Study

As mentioned in Section 4.3, our CausalDiff contains Adversarial Purification (AP), Causal Factor Inference (CFI), and Latent-S-Based Classification (LSBC). The last block of Table 2 shows the individual effect of the AP and CFI in CausalDiff. We can see: 1) Our DDPM-based purification (CausalDiff w/o CFI, achieved by AP plus a standard classifier) performs similarly to LM-EDM and is significantly better than LM-DDPM, showing that the strategy of sampling within small timesteps for purification is much more effective than entire timesteps. (We show more analysis on this in Appendix C.2. ) 2) The core component of CasualDiff - causal disentanglement (CausalDiff w/o AP) outperforms all the baselines in terms of average robustness except RDC which incorporates an extra LM purification step. It shows that modeling the generative of native in-domain data can enhance the model's inherent robustness and thus effectively defend against various types of attacks.

## 5.4 Visualization of Latent Factors

To understand how the latent causal factors $S$ and $Z$ in CausalDiff take effect during adversarial classification, we visualize $S$, $Z$, and their concatenation using t-SNE in Fig. 4. We randomly sampled 5000 correctly classified test samples from CIFAR-10 for visualization. We find that the Y-causative factor $S$ of samples in each category are located in the same cluster, with clear margins between different clusters except the categories - dog, cat, and bird. These three categories share more commonalities than the others and are not surprising to have blurred boundaries. Additionally,

the $S$ vectors of airplanes (dark blue) are near those of birds (green), and trucks have $S$ vectors near automobiles. In contrast to the $S$ vectors, the vectors of $Z$ do not exhibit correlations with the categories. This also aligns with our objective to extract the Y-non-causative factors to $Z$. The vectors of their concatenation, i.e., $[s; z]$, also display in clusters but with much more blurred boundaries. These observations are consistent with our commonsense knowledge, showing that CausalDiff has learned reasonable Y-causative factors by $S$ and Y-non-causative factors by $Z$.

## 6 Conclusion

We develop a causal model based on diffusion model to improve adversarial robustness. A pilot study on toy data suggests that the model defends against adversarial attacks by leveraging label-causative features to resist perturbations and expand the model's margin. Moreover, on the CIFAR-10, CIFAR-100 and GTSRB datasets, our model appears to capture semantic features consistent with the human decision-making process and surpass all baseline models, achieving state-of-the-art performance in adversarial robustness, particularly against unseen attacks.

## Acknowledgments and Disclosure of Funding

This work is supported by the Strategic Priority Research Program of the Chinese Academy of Sciences, Grant No. XDB0680101, CAS Project for Young Scientists in Basic Research under Grant No. YSBR-034, the Innovation Project of ICT CAS under Grant No. E261090, the National Natural Science Foundation of China (NSFC) under Grants No. 62302486, the Innovation Project of ICT CAS under Grants No. E361140, the CAS Special Research Assistant Funding Project, the project under Grants No. JCKY2022130C039, and the Strategic Priority Research Program of the CAS under Grants No. XDB0680102. We would like to thank the UIBE GVC Laboratory and the Digital Economy Laboratory at the University of International Business and Economics for their provision of computational resources and support.

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

# A Proof of Propositions

In this section, we will present the detailed proof for the theoretical results mentioned in the main paper.

## A.1 Proof of Causal Information Bottleneck (CIB)

According to the Structural Causal Model (SCM), we have $p(x, y, s, z) = p(s)p(z)p(x|s, z)p(y|s)$. Thus, the Causal Information Bottleneck (CIB) can be represented as:

$$
\begin{aligned}
\mathrm{I}(X, Y; S, Z) &= \mathbb{E}_{p(x,y,s,z)} \log \frac{p(x, y, s, z)}{p(x, y)p(s, z)} \\
&= \mathbb{E}_{p(x,y,s,z)} \log \frac{p(s)p(z)p(x|s, z)p(y|s)}{p(x, y)p(s, z)} \\
&= \mathbb{E}_{p(x,y,s,z)} \log \frac{p(x|s, z)p(y|s)}{p(x, y)} + \mathbb{E}_{p(x,y,s,z)} \log \frac{p(s)p(z)}{p(s, z)} \\
&= \mathbb{E}_{p(x,y,s,z)} \log \frac{p(x|s, z)p(y|s)}{p(x, y)} - \mathrm{I}(S; Z) \\
&= \mathbb{E}_{p(x,y,s,z)} \log \frac{p(x|s, z)p(y|s)p(s, z)}{p(y|x)p(x)p(s, z)} - \mathrm{I}(S; Z) \\
&= \mathbb{E}_{p(x,y,s,z)} \log \frac{p(y|s)}{p(y|x)} + \mathbb{E}_{p(x,y,s,z)} \log \frac{p(x|s, z)p(s, z)}{p(x)p(s, z)} - \mathrm{I}(S; Z) \\
&= \mathbb{E}_{p(x,y,s,z)} \log \frac{p(y|s)}{p(y|x)} + \mathrm{I}(X; S, Z) - \mathrm{I}(S; Z) \\
&= \mathbb{E}_{p(x,y,s,z)} \log \frac{p(y|s)p(s)p(y)}{p(y|x)p(s)p(y)} + \mathrm{I}(X; S, Z) - \mathrm{I}(S; Z) \\
&= \mathbb{E}_{p(x,y,s,z)} \log \frac{p(y)}{p(y|x)} + \mathbb{E}_{p(x,y,s,z)} \log \frac{p(y|s)p(s)}{p(s)p(y)} + \mathrm{I}(X; S, Z) - \mathrm{I}(S; Z) \\
&= \mathbb{E}_{p(x,y,s,z)} \log \frac{p(y)p(x)}{p(y|x)p(x)} + \mathrm{I}(Y; S) + \mathrm{I}(X; S, Z) - \mathrm{I}(S; Z) \\
&= \mathrm{I}(X; Y) + \mathrm{I}(Y; S) + \mathrm{I}(X; S, Z) - \mathrm{I}(S; Z)
\end{aligned}
\tag{11}
$$

Therefore, we have proved the result in Eq. (5)

## A.2 Proof of the Lower Bound of CIB

According to the result in Eq. (11), we further prove its lower bound shown in Eq. (6) in this section.

As for the lower bound of the recontruction term $\mathrm{I}(X; S, Z)$, we have:

$$
\begin{aligned}
\mathrm{I}(X; S, Z) &= \mathbb{E}_{p(x,s,z)}[\log \frac{p(x|s, z)}{p(x)}] \\
&= \mathbb{E}_{p(x,s,z)}[\log \frac{p(x|s, z)p_\theta(x|s, z)}{p(x)p_\theta(x|s, z)}], \quad \text{when using a variational distribution } p_\theta(x|s, z) \text{ to approximate } p(x|s, z), \\
&= \mathbb{E}_{p(x,s,z)}[\log \frac{p_\theta(x|s, z)}{p(x)}] + \mathbb{E}_{p(s,z)}\mathbb{E}_{p(x|s,z)}[\log \frac{p(x|s, z)}{p_\theta(x|s, z)}] \\
&= \mathbb{E}_{p(x,s,z)}[\log \frac{p_\theta(x|s, z)}{p(x)}] + \mathbb{E}_{p(s,z)}[\mathcal{D}_{\mathrm{KL}} \frac{p(x|s, z)}{p_\theta(x|s, z)}], \quad \text{where } \mathcal{D}_{\mathrm{KL}}(\cdot) \text{ represents KL-divergence,} \\
&\geq \mathbb{E}_{p(x,s,z)}[\log \frac{p_\theta(x|s, z)}{p(x)}] \\
&= \mathbb{E}_{p(x,s,z)}[\log p_\theta(x|s, z)] + \mathbb{E}_{p(x,s,z)}[\log \frac{1}{p(x)}] \\
&= \mathbb{E}_{p(x,s,z)}[\log p_\theta(x|s, z)] + \mathcal{H}(x), \quad \text{where } \mathcal{H}(x) \text{ indicates entropy of } x.
\end{aligned}
\tag{12}
$$

As for the upper bound of $I(X; S, Z)$, we have:

$$
\begin{aligned}
I(X; S, Z) &= \mathbb{E}_{p(x,s,z)}[\log \frac{p(x|s,z)}{p(x)}] \\
&= \mathbb{E}_{p(x,s,z)}[\log \frac{p(x|s,z)q(s,z)}{p(x)q(s,z)}], \text{ when using a prior distribution } q(s,z) \text{ to estimate } p(s,z), \\
&= \mathbb{E}_{p(x,s,z)}[\log \frac{p(x|s,z)}{q(s,z)}] - \mathcal{D}_{\mathrm{KL}}(p(s,z)\|q(s,z)), \text{ where } \mathcal{D}_{\mathrm{KL}}(\cdot) \text{ represents KL-divergence,} \\
&\leq \mathbb{E}_{p(x,s,z)}[\log \frac{p(x|s,z)}{q(s,z)}] \\
&= \mathbb{E}_{p(x)}\mathbb{E}_{p(s,z|x)}[\log \frac{p(s,z|x)}{q(s,z)}] \\
&= \mathbb{E}_{p(x)}[\mathcal{D}_{\mathrm{KL}}(p(s,z|x)\|q(s,z))].
\end{aligned}
\tag{13}
$$

Regarding the label prediction term $I(Y; S)$, we can maximize the mutual information between factor $S$ and label $Y$ by maximize $\mathbb{E}_{p(y,s)}[\log p_\theta(y|s)]$ as well as employing a cross-entropy loss function, according to Boudiaf et al. [63].

As for the disentangle term $I(S; Z)$, according to the Contrastive Log-Ratio Upper Bound (CLUB) of mutual information proposed by Cheng et al. [57], we have

$$
I(S; Z) \leq I_{\mathrm{CLUB}}^\theta(S; Z) = \mathbb{E}_{p(s,z)}[\log p_\theta(s|z)] - \mathbb{E}_{p(z)}\mathbb{E}_{p(s)}[\log p_\theta(s|z)],
\tag{14}
$$

where $p_\theta(s|z)$ is a variational distribution to estimate $p(s|z)$.

Thus, we have proved the results in Eq. (6) that

$$
\begin{aligned}
&I(X; S, Z) + I(Y; S) - I(S; Z) - \lambda I(X; S, Z) \\
&\geq \mathbb{E}_{p(x,s,z)}[\log p_\theta(x|s,z)] + \mathbb{E}_{p(y,s)}[\log p_\theta(y|s)] - I_{\mathrm{CLUB}}^\theta(S; Z) - \lambda \, \mathbb{E}_{p(x)}[\mathcal{D}_{\mathrm{KL}}(p_\theta(s,z|x)\|q(s,z))]. \\
&= \mathbb{E}_{p(x,s,z)}[\log p_\theta(x|s,z)] + \mathbb{E}_{p(y,s)}[\log p_\theta(y|s)] - \mathbb{E}_{p(s,z)}[\log p_\theta(s|z)] \\
&\quad + \mathbb{E}_{p(z)}\mathbb{E}_{p(s)}[\log p_\theta(s|z)] - \lambda \, \mathbb{E}_{p(x)}[\mathcal{D}_{\mathrm{KL}}(p_\theta(s,z|x)\|q(s,z))].
\end{aligned}
\tag{15}
$$

### A.3 Detailed Derivation of Loss Function

Based on the result in Eq. (15), we can optimize the Causal Information Bottleneck (CIB) $I(X, Y; S, Z)$ by maximizing its theoratically lower bound. In this part, we discuss the detailed derivation for the lower bound of CIB in term of designing the loss function proposed in Eq. (7).

Specifically, for the reconstruction term $\mathbb{E}_{p(x,s,z)}[\log p_\theta(x|s,z)]$, we can maximize the log-likelihood estimated by our conditional diffusion model:

$$
\log p_\theta(x|s,z) \geq -\mathbb{E}_{\epsilon,t}[w_t\|\epsilon_\theta(x_t, t, s, z) - \epsilon\|] + C,
\tag{16}
$$

where $\epsilon_\theta(x_t, t, s, z)$ denotes our conditional diffusion model, and constant $C$ is negligible [17].

Regarding the label prediction term $\mathbb{E}_{p(y,s)}[\log p_\theta(y|s)]$, following the results proposed by Boudiaf et al. [63], we can leveraging a cross-entropy loss function to maximize the mutual information between factor $S$ and label $Y$.

As for the disentanglement term $I(S; Z)$, we following the optimization strategies proposed by Cheng et al. [57], leveraging a predictor $p_\theta$ to learn the relationship between $S$ and $Z$ so as to estimate $I_{\mathrm{CLUB}}^\theta(S; Z)$.

Overall, we have proved the loss function of our Causal Information Bottleneck (CIB) proposed in Eq. (7):

$$
\begin{aligned}
\mathcal{L}(x, y, s, z; \theta) ={}& \alpha \, \mathbb{E}_{\epsilon,t}\|\epsilon_\theta(x_t, t, s, z) - \epsilon_t\|_2^2 + \gamma \mathcal{L}_{\mathrm{CE}}(s, y; \theta) \\
&+ \eta\{\mathbb{E}_{p(s,z)}[\log p_\theta(s|z)] - \mathbb{E}_{p(z)}\mathbb{E}_{p(s)}[\log p_\theta(s|z)]\} + \lambda \, \mathcal{D}_{\mathrm{KL}}(p_\theta(s,z|x)\|q(s,z)),
\end{aligned}
\tag{17}
$$

where $\mathbb{E}_{p(s,z)}[\log p_\theta(s|z)] - \mathbb{E}_{p(z)}\mathbb{E}_{p(s)}[\log p_\theta(s|z)]$ is the estimation of $I_{\mathrm{CLUB}}^\theta(S; Z)$.

---

**Algorithm 1** CausalDiff Algorithm

---

**Require:** Dataset $\mathcal{D}$; The CausalDiff parameterized by $\theta$ pretrained by Algorithm 2 involves an UNet $\epsilon_\theta(x_t, t, s, z)$ with diffusion timestep $T$, an encoder $f_{(s,z)}(x; \theta)$, a classifier $f_y(s; \theta)$, and an MI estimator $f_{\text{CLUB}}(s, z; \theta)$ for the CLUB loss; an optimizer $\text{optim}(\cdot)$, dropout probability $p_{\text{drop}}$ of $s, z$ for simultaneously training an unconditional generation, hyperparameters $\alpha, \lambda, \gamma, \eta$.

---

**Initialize:**
    Load $f_{s,z}(x; \theta)$ and $\epsilon_\theta$ from the pretrained model $\theta$.
    Initialize $f_y(s; \theta)$ and $f_{\text{CLUB}}(s, z; \theta)$ randomly.
**for** ep=1 **to** $N_2$ **do**
    Get mini-batch $(x, y) \sim \mathcal{D}$
    $s, z = f_{(s,z)}(x; \theta)$
    $t \sim \text{Uniform}(\{1, 2, ..., T\}), \epsilon \sim \mathcal{N}(\mathbf{0}, \mathbf{I})$
    **if** $rand(0, 1) \leq p_{\text{drop}}$ **then**
        Compute loss $\mathcal{L}(x; \theta) = \|\epsilon_\theta(x_t, t) - \epsilon\|_2^2$
        Update $\epsilon_\theta$ with $\text{optim}(\theta, \mathcal{L}(x; \theta))$
    **else**
        Compute loss $\mathcal{L}(x, y, s, z; \theta)$ according to Eq. (7)
        Update $\epsilon_\theta(x_t, t, s, z), f_{(s,z)}(x; \theta),$
            and $f_y(s; \theta)$ by $\text{optim}(\theta, \mathcal{L}(x, y, s, z; \theta))$
    **end if**
    Update $f_{\text{CLUB}}(s, z; \theta)$ according to the CLUB algorithm [57]
**end for**

---

### A.4 ELBO (Evidence Lower BOund) V.S. MI (Mutual Information)

the Causal Evidence Lower BOund (ELBO) for multi-domain datasets as proposed by Sun et al. [14], we directly formulate the Causal ELBO within our causal structure, as depicted in Fig. 1 (Left). Given that $p(x, y, s, z) = p(s)p(z)p(x|s, z)p(y|s)$, we can derive the Causal ELBO in single domain as follows:

$$
\begin{aligned}
\text{ELBO} &= \mathbb{E}_{p(\boldsymbol{x}, \boldsymbol{y})}[\mathbb{E}_{q_\psi(s, z|x, y)} \log \frac{p_\theta(x, y, s, z)}{q_\psi(s, z|x, y)}] \\
&= \mathbb{E}_{p(x, y)} \Big\{ \mathbb{E}_{q_\psi(s, z|x, y)} \Big[ \log p_\theta(x \mid s, z) + \log \frac{p_\theta(s, z)}{q_\psi(s, z \mid x)} + \log p_\theta(y \mid s) + \log \frac{q_\psi(y \mid x)}{q_\psi(y \mid s)} \Big] \Big\},
\end{aligned}
\tag{18}
$$

where $p_\theta$ is to learn the ground-truth $p$ and $q_\psi$ is variational distribution to mimic $p_\theta$.

Specifically, the causal ELBO, compared to our Causal Information Bottleneck (CIB), incorporates the reconstruction term $\log p_\theta(x \mid s, z)$, the insensitivity term $\log \frac{p_\theta(s, z)}{q_\psi(s, z|x)}$, and the label prediction term $\log p_\theta(y \mid s)$. However, it overlooks the disentanglement of latent factors, a critical aspect for effectively learning the causal model. Additionally, we have empirically assessed the robustness of models trained with either CIB or causal ELBO (equivalent to $\eta = 0$) in section C.1. This evaluation aims to investigate the efficacy of the disentanglement term $\text{I}(S; Z)$.

## B   More Implementation Details

### B.1   Baselines

We include representative defense methods of adversarial training (AT), purification, and other types (e.g., generative-model-based approach) as baselines. Specifically, we compare with the AT methods [9, 23] that use DDPM or the Elucidating Diffusion Model (EDM) [49] for data augmentation and a theoretical framework TRADES [20] for adversarial training. We also include causality-based AT baselines such as CausalAdv [15] and DICE [16]. Purification baselines include DiffPure [33] (grounded on Score SDE [30]) and diffusion-based Likelihood Maximization (LM)[27] with EDM (as in the original paper) and DDPM (we reproduced). Other defense baselines comprise generative classifiers such as Score-Based Generative Classifier (SBGC) [59], [27], and the causality-based

---

**Algorithm 2** CausalDiff Pretrain Algorithm

---

**Require:** Dataset $\mathcal{D}$; a diffusion model $\epsilon_\theta$, an encoder $f_{(s,z)}(x;\theta)$, a classifier $f_y(s;\theta)$; an optimizer $\text{optim}(\cdot)$, probability $p_{\text{drop}}$ of training samples for unconditional diffusion, diffusion training epoch $N_1$.

---

  **Initialize:** randomly initialize parameter $\theta$.
  **for** ep=1 **to** $N_1$ **do**
    $(x, y) \sim \mathcal{D}$
    $s, z = f_{(s,z)}(x; \theta)$
    $s, z \leftarrow \phi$ with probability $p_{\text{drop}}$     ▷ randomly mask the latent factors with probability $p_{\text{drop}}$
    $t \sim \text{Uniform}(\{1, 2, ..., T\})$, $\epsilon \sim \mathcal{N}(\mathbf{0}, \mathbf{I})$
    $\mathcal{L}(x, s, z; \theta) = \|\epsilon_\theta(x_t, t, s, z) - \epsilon\|_2^2$
    Update $\epsilon_\theta$ and $f_{(s,z)}(x; \theta)$ by $\text{optim}(\theta, \mathcal{L}(x, s, z; \theta))$
  **end for**

---

---

**Algorithm 3** Adversarially Robust Inference Algorithm

---

**Require:** A fully-trained causal model involves diffusion model $\epsilon_\theta(x_t, t, s, z)$, encoder $f_{(s,z)}(x; \theta)$, and classifier $f_y(s; \theta)$; test image $x$, purification optimization steps $N_{\text{purify}}$, causal factor inference steps $N_{\text{infer}}$, number of sampling steps $N_t$ for inference, optimizer $\text{optim}_{\text{purify}}$ for purification, optimizer $\text{optim}_{\text{infer}}$ for causal factor inference.

---

  **Initialize:** purified image $x^* = x$
  **Stage 1: Adversarial Purification**
  **for** iter=1 **to** $N_{\text{purify}}$ **do**
    $t \sim \text{Uniform}(\{1, 2, ..., 50\})$, $\epsilon \sim \mathcal{N}(\mathbf{0}, \mathbf{I})$
    $\mathcal{L}(x^*, \theta) = \|\epsilon_\theta(x_t, t) - \epsilon\|_2^2$
    Update $x^*$ by $\text{optim}_{\text{purify}}(x^*; \mathcal{L}(x^*, \theta))$
  **end for**

  **Stage 2: Causal Factor Inference**
  Initial $s^*, z^* = f_{(s,z)}(x^*; \theta)$
  **for** iter=1 **to** $N_{\text{infer}}$ **do**
    Sample $N_t$ timesteps $t$ at equal intervals from 1 to $T$
    Sample $N_t$ $\epsilon \sim \mathcal{N}(\mathbf{0}, \mathbf{I})$ corresponds to each $t$
    Compute $\mathcal{L}(x^*, s^*, z^*; \theta) = \mathbb{E}_{\epsilon, t} \|\epsilon_\theta(x_t^*, t, s, z) - \epsilon\|_2^2$
    Update $s^*, z^*$ by $\text{optim}_{\text{infer}}(s^*, z^*; \mathcal{L}(x^*, s^*, z^*; \theta))$
  **end for**

  **Stage 3: S-based Classification**
  $y = f_y(s^*; \theta)$
  **return** $y$

---

generative model CAMA [43]. Notably, we also include the SOTA method on unseen attacks - Robust Diffusion Classifier (RDC), which incorporates purification and conditional generation based on labels for defense.

## B.2 Implementation Details

We use DDPM [17] as our generative model. For the encoder $f_{s,z}(x; \theta)$ and the classification model $f_s(y; \theta)$, we employ WideResNet-70-16 (WRN-70-16) as the backbone.

**Training Strategy.** Considering the different complexities in learning $f_x(s, z; \theta)$, $f_{(s,z)}(x; \theta)$ and $f_y(s; \theta)$, attributed to the differing difficulty in generation and discrimination task, we segment the training of the entire causal model into two distinct phases. As outlined in Section 4.2, we employ a two-stage training process for our CausalDiff.

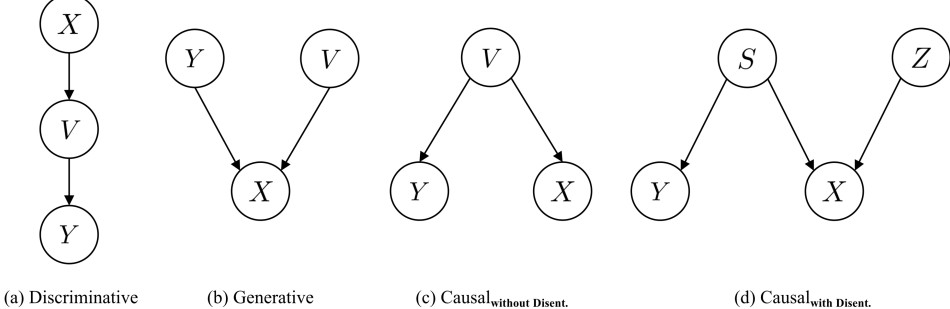

Figure 5: SCM of models for pilot study including (a) discriminative model, (b) generative model, (c) causal model without disentanglement, and (d) causal model with disentanglement.

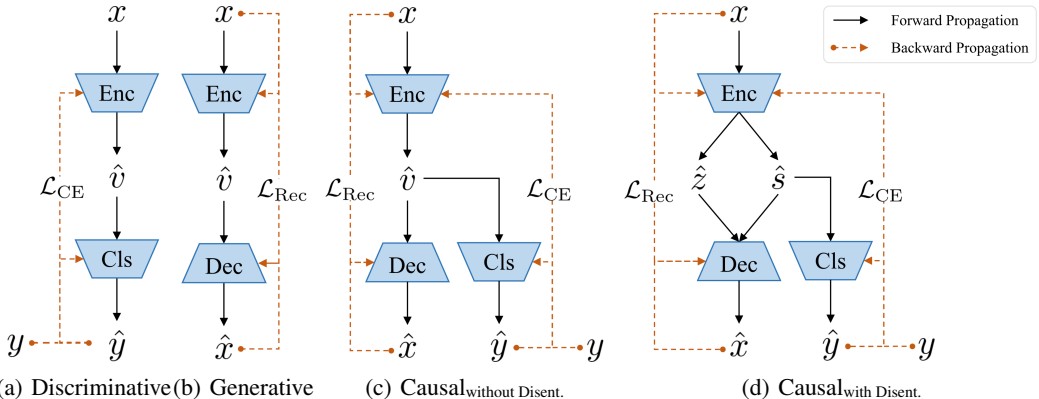

Figure 6: Architecture of models for pilot study including (a) discriminative model, (b) generative model, (c) causal model without disentanglement, and (d) causal model with disentanglement.

In the pretrain phase on CIFAR-10 and GTSRB datasets, focused on generation, primarily trains the conditional diffusion model $f_x(s, z; \theta)$ along with its corresponding encoder $f_{(s,z)}(x; \theta)$ according to Algorithm 2, setting $N_1 = 1440$. For CIFAR-100, we added 10,000 images generated by EDM [49] to our training set. Considering the computational cost, we only pretrain for $N_1 = 500$ epochs. Note that the augmented data is used only during the pretraining phase while not in the joint training phase.

Subsequently, we conduct joint training of the whole CausalDiff model for $N_2 = 560$, amounting to a total of 2000 epochs. The second phase, targeting discrimination and leveraging label information to guide disentanglement, involves the joint training of the entire causal model. Note that, in order to simultaneously train an unconditional diffusion model for adversarial purification, we follow Ho and Salimans [51] to mask the condition $s$ and $z$ with probability $p_{\text{drop}} = 0.1$. Thus, a single shared model is used for both adversarial purification and causal factor inference.

Both the pretraining and joint training phases utilize a learning rate of $1e^{-4}$ and a batch size of 128. For simplicity, we follow the setting of $w_t = 1$ [17]. We set $\alpha = 1., \gamma = 1e^{-2}, \eta = 1e^{-5}, \lambda = 1e^{-2}$ as the weights for the loss function in Eq. (7).

Since we need a standard diffusion model $\epsilon_\theta(x_t, t)$ for purification during adversarial inference, we apply dropout of $s$ and $z$ with a ratio of $p_{\text{drop}} = 0.1$ for conditional diffusion generation as in Ho et al. [17] during both pretraining and training. Thus, the unconditional probability of generating $x$ can also be estimated using the same model by masking $s$ and $z$.

**Inference Strategy.** Leveraging the trained CausalDiff, we can infer its label from an adversarial example in accordance with Algorithm 3, following the inference pipeline outlined in Section 4.3. Specifically, we set $N_{\text{purify}} = 5$ and use momentum-SGD as our $\text{optim}_{\text{purify}}$, with a learning rate of

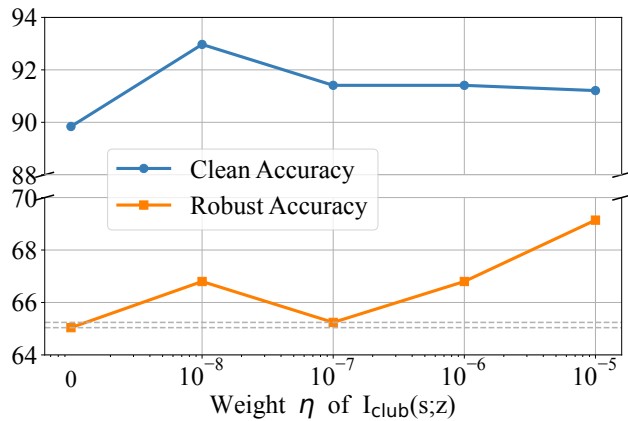

Figure 7: Impact of $\eta$ for disentanglement term in loss function on clean accuracy and robust accuracy.

0.1. For causal factor inference, we choose $N_t = 10$ to sample 10 timesteps per iteration and adopt $N_{\text{infer}} = 10$ with momentum-SGD as $\text{optim}_{\text{infer}}$, setting the learning rate to $1e^{-5}$.

Regarding white-box attacks, we perform a gradient backpropagation throughout the entire pipeline of our CausalDiff approach, which includes purification, causal factors inference, and classification. This implies that the attacker possesses sufficient knowledge of the causal model.

**Attack Evaluation.** Attack evaluation for CIFAR-10 dataset includes a 100-step StAdv attack [60] with $\epsilon = 0.05$ under the $\ell_\infty$ norm bound, BPDA+EOT (EOT=20) against the $\ell_\infty$ threat model with $\epsilon = 8/255$, and AutoAttack [2] (AA), which comprises 100-step white-box attacks such as APGD-ce, APGD-t, FAB-t, and a 5000-step black-box Square Attack under both $\ell_2$ ($\epsilon = 0.5$) and $\ell_\infty$ ($\epsilon = 8/255$) constraints.

For the GTSRB dataset, we utilize **four types of attacks as well as two types of corruptions** to evaluate robustness against adversarial attacks and the influence of the natural environment. Specifically, the attack methods include AutoAttack, which comprises 100-step white-box attacks such as APGD-ce, APGD-t, FAB-t, and a 5000-step black-box Square Attack under both $\ell_2$ ($\epsilon = 0.5$) and $\ell_\infty$ ($\epsilon = 8/255$) constraints. The corruptions include Fog and Brightness. Following Nie et al. [33], we randomly sample 512 samples for evaluation.

### B.3 Details for Pilot Study

**Data Construction.** For each data point of the 2000 samples, we sample a vector $s$ with dimension $h_s = 8$ from a normal distribution with mean $-1$ and variance 1, i.e., $s \sim \mathcal{N}(-1, 1)$, and a vector $z \sim \mathcal{N}(1, 1)$ with dimension $h_z = 8$. Subsequently, we projected the concatenation of $s$ and $z$, denoted as $[s; z]$, to a sample $x$, with a random initialized matrix $A_x(A_x \in \mathcal{R}^{(h_s + h_z) \times h_x})$, i.e., $x = [s; z] \cdot A_x$. Similarly, we produced the score $y_s$ of $x$ with $y_s = s \cdot A_y$, where $A_y \in \mathcal{R}^{h_s \times 1}$. To obtain samples with balanced binary labels, we consider the label $y$ of the sample with $y_s$ above the median as 1 and the others as 0.

**Methods for Comparisons.** In the pilot study detailed in Section 3, we conducted an investigation and analysis on four models: 1) Discriminative: a discriminative model that learns to classify the samples with a two-layer perceptron (MLP), 2) Generative [27]: a generative model that learns the generation of the sample $x$ conditioning on its label $y$ and predict the label of an adversarial example by calculating the $\max_y p(x|y)$, 3) Causal without Disent.: a causal model that models the generation of both x and y with the same causal factor $v$ (Causal modeling without Disentanglement), 4) Causal with Disent.: our model that disentangles the label-causative factor $s$ and another factor $z$ during the generation of $x$. For the latter two causal models, given an adversarial example, the hidden vectors $v$ or $s, z$ are inferred for Causal without Disent. and with Disent. which are then used for the final label prediction. This section comprehensively presents the designed causal structures in Fig. 5 and the model architectures in Fig. 6.

Regarding implementation, we trained each of the four models for 20 epochs, optimizing the model parameters using the Adam optimizer with a learning rate of $1e^{-3}$. The latent dimension for each model was set to 64. For evaluation, we employed a 100-step PGD attack with $\epsilon = 0.3$ and $\alpha = 2/255$ within the $\ell_\infty$ norm boundary. The variation in latent variables and predicted logits between adversarial examples and clean images, as presented in Table 4, is measured on adversarial examples generated with $\epsilon = 10$ and $\alpha = 0.05$ within the $\ell_\infty$ norm boundary, using a 100-step PGD attack.

## C    More Experimental Results

### C.1    Analyses on Core Components of Training

**Effect of** $\mathrm{I}^\theta_{\mathrm{CLUB}}(S; Z)$ **in Casual Information Bottleneck (CIB)** To examine the impact of our introduced disentanglement term $\mathrm{I}^\theta_{\mathrm{CLUB}}(S; Z)$ in CIB (See Equation (6)), we vary $\eta$ in Equation (7) and evaluate the robustness against the most challenging attack in AutoAttack [2], i.e., with $\ell_\infty$ norm bounded by $\epsilon = 8/255$. Larger $\eta$ will cause the diffusion model to collapse, so we do not include the results of larger $\eta$. As we mentioned in Section 4.2, our CIB regresses to the ELBO objective in [14] when $\eta = 0$. As shown in Fig. 7, CausalDiff has better clean accuracy as well as robustness when $\eta > 0$ and yields the best robustness when $\eta = 10^{-5}$ (69.14% compared to 65.04% when $\eta = 0$). It confirms that $\mathrm{I}^\theta_{\mathrm{CLUB}}(S; Z)$ in the loss function is beneficial to disentangle the Y-causative from Y-non-causative factors and can further enhance both clean accuracy and robustness.

### C.2    Analyses on Core Components of Inference

**Causal Factor Inference Method**. We also evaluate the robustness using the encoder $f_{(s,z)}(x; \theta)$ to get latent variables instead of inferring $s$ and $z$ through the conditional diffusion model. The results reveal that classification by the encoder (without purification) achieves a clean accuracy of 91.99% but 0.00% against AutoAttack under both $\ell_\infty$ and $\ell_2$ threat models. This decline might be attributed to imprecise modeling around $x$ (i.e., $x + \delta$), which results in an inability to resist adversarial perturbations on $x$.

**Timestep $t$ Sampling Strategies for Purification** As discussed in Section 5.2, our purification method (*CausalDiff_{w/o Causal Factor Inference}* in Table 2) markedly surpasses the direct adaptation of likelihood maximization (*LM-DDPM* in Table 2), as proposed by Chen et al. [27], applied to DDPM. This improvement stems from a refined strategy in sampling the timestep $t$.

As demonstrated in Fig. 9, we found that smaller timesteps perform better in distinguishing between the distributions of clean and adversarial samples. Specifically, we presented the negative log-likelihood estimated by expectation $\mathbb{E}_\epsilon[w_t\|\epsilon_\theta(x_t, t) - \epsilon\|]$ for the given timestep over 512 examples. This may caused by a larger timestep implies a greater degree of noise addition for estimating likelihood, which might overshadow the adversarial perturbations, unexpected for purification.

### C.3    Visualization of Cases

We visualize the generated images leveraging the conditional generation of our CausalDiff, providing an intuitive depiction of the label-causative factor $s^*$ and the label-non-causative factor $z^*$. Fig. 8 illustrates that after inferring from the benign example $x^*$, perturbations are alleviated. The image $x_{s^*}$ conditioned on $s^*$ showcases that $s^*$ captures the core predictive features, reflecting the general concept of the category (for instance, a common semantic representation of 'horse' from an image of a brown horse's head), or even enhances the predictive features such as the dog's face or the ship's body, whereas $z^*$ retains specific and non-predictive image details.

We also visualize the purified example $x^*$, conditionally generated $x^*_{s^*}$ and $x^*_{z^*}$ when encountering an adversarial example $\tilde{x}$. These cases demonstrate that, despite the presence of perturbations in the clean images, our CausalDiff effectively captures the correct predictive information of $s^*$, maintaining alignment with the clean data.

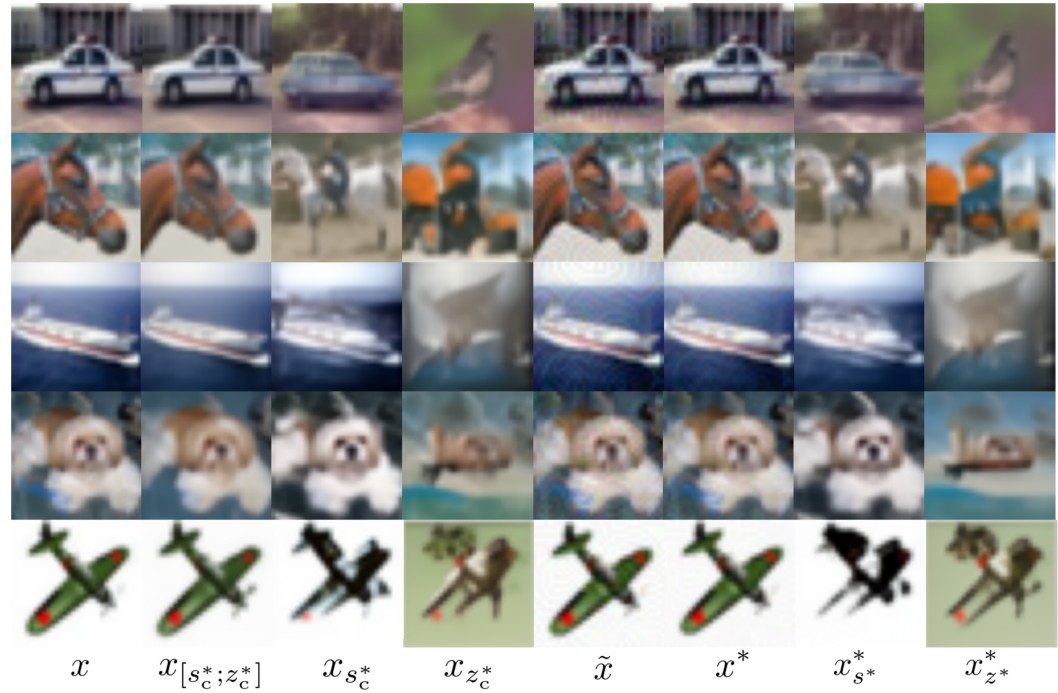

$$x \qquad x_{[s_c^*; z_c^*]} \qquad x_{s_c^*} \qquad x_{z_c^*} \qquad \tilde{x} \qquad x^* \qquad x_{s^*}^* \qquad x_{z^*}^*$$

Figure 8: Reconstruction images $x_{[s_c^*; z_c^*]}$ when given clean example $x$, where $s_c^*$ and $z_c^*$ are inferred from the clean example $x$ by our CausalDiff; generated images $x_{s^*}$ and $x_{z^*}$ conditioned on $s_c^*$ and $z_c^*$, respectively; purified image $x^*$ utilizing the unconditional diffusion (with $s, z$ masked) when given an adversarial example $\tilde{x}$; generated images $x_{s^*}$ and $x_{z^*}$ conditioned on $s^*$ and $z^*$, respectively, where $s^*$ and $z^*$ are inferred from the purified image $x^*$.

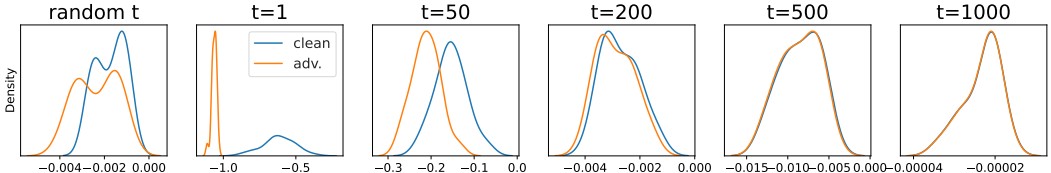

Figure 9: Distribution of likelihood for adversarial and benign examples across various timesteps $t$.

## C.4 Speed Test of Inference Time

We evaluate the computational complexity of CausalDiff and DiffPure [33] as well as a discriminative model (WRN-70-16) by measuring the inference time in seconds for a single sample (average on 100 examples from CIFAR-10 dataset) on two types of GPUs, including NVIDIA A6000 GPU and 4090 GPU (Our experiments leverage 4 A6000 GPUs and 4 4090 GPUs). The results are shown in Table 5.

Table 5: Comparison of NFEs (Number of Function Evaluations) across different models on GPUs

|  | CausalDiff | CausalDiff w/o Purify | CausalDiff w/o Causal Factor Infer. | DiffPure | WRN-70-16 |
|---|---|---|---|---|---|
| NFE | $N_1 + N_2 + 1$ | $N_2 + 1$ | $N_1 + 1$ | $N_1$ | 1 |
| Time (A6000) | 4.97 | 4.62 | 0.29 | 2.22 | 0.011 |
| Time (4090) | 4.88 | 4.61 | 0.25 | 2.06 | 0.007 |

# D Limitation

Although our CausalDiff significantly narrows the gap in classification accuracy between adversarial and clean examples, it requires an inference cost of $1 + N_1 + N_2$ NFEs (Number of Function Evaluations), where efficiency improvements are needed. Note that $N_1$ indicates the purification step (e.g. 5) and $N_2$ indicates the step of causal factor inference (e.g., 10) and 1 NFE for latent-S-based classification. Furthermore, our CausalDiff represents a new framework, meaning it requires training from scratch. Perhaps in the future, an efficient implementation of robust inference could be achieved by embedding causal mechanisms into the existing models in a plug-and-play manner.

# E Broader Impact

Our CausalDiff model, built upon a powerful generative framework, aims to align with human decision-making mechanisms to enhance the stability and trustworthiness of neural networks. This approach holds potential for advancing the field of Machine Learning, particularly in safety-sensitive applications such as autonomous driving and facial recognition.

