# OpenReview forum: "CausalDiff: Causality-Inspired Disentanglement via Diffusion Model for Adversarial Defense"
_NeurIPS.cc/2024/Conference — NeurIPS 2024 poster_

### Official Review · Reviewer_pU16 · 2024-07-08

**Soundness:** 3
**Presentation:** 3
**Contribution:** 3
**Rating:** 6
**Confidence:** 4

**Summary:**

The paper introduces an adversarial defense method, CausalDiff, which leverages a causal model of robustness combined with a diffusion model to learn label-relevant causative representations. The effectiveness of CausalDiff is specifically demonstrated against unseen attacks, where it surpasses existing adversarial defense baselines on several image classification datasets.

**Strengths:**

- This paper is well-motivated, presenting a novel framework that extracts the disentangled essential features of an image, achieved by first training a diffusion model and then inferring from it.
- The pilot study is easy to understand, and the results persuasively demonstrate the effectiveness of the proposed causality-inspired defense framework.
- A comprehensive comparison with baselines of adversarial defense methods, especially those based on Diffusion Models, makes the results convincing.

**Weaknesses:**

- Limited discussion about related works. To my knowledgement, neither CausalAdv and DICE limits their modeling of adversarial distribution to a certain type of attack. Therefore, it is necessary to discuss how the proposed method differs from these important baselines in terms of causal modeling.
- The writing can be improved. The challenges listed from line 53 to 66 could be clearer by focusing on highlighting the actual problems that are solved by the proposed method, rather than just listing how the method is implemented. For example, challenge (2) could be clarified by explaining that applying a *Causal Information Bottleneck* objective aims to minimize the essential factors to their minimal necessary extent.
- The empirical results of the pilot study are convincing. However, the insight into how to extract the desired features (i.e., S, Z) appears unclear, as detailed in Q2-Q3.

**Questions:**

Q1: Could you provide further insight into the source of CausalDiff’s ability to generalize to unseen attacks? For example, what attributes of the *essential factor* for benign images enable its generalizability to unseen attacks, and are there any specific constraints regarding the types of unseen attacks?

Q2: Could the author further explain the reason for the opposite optimization direction $(1 - \lambda) I(X;S,Z)$ in Eq. 5? E.g., what is the relationship between the motivation of avoiding "X containing too many unimportant details" (line 196) and the "insensitivity" for $-\lambda I(X;S,Z)$ in Fig. 2? These statements seem to imply the existence of a feature that is neither "essential" (i.e., S) nor "non-essential but important for reconstructing X" (i.e., Z). This kind of feature appears crucial for "sensitivity" but is not fully discussed in the paper.

Q3: Any abation study on the parameter $\lambda$ used in Eq.5, which typically plays a crucial role in IB-style objectives?

Q4: Other minor typos.

- There is a citation format error in line 21, and a method name mistake (with / without) in line 124.

**Limitations:**

- The defense performance is superior against adversarial attacks with  L_p  norm constraints. However, the effectiveness of the proposed method against unbounded attacks (e.g., semantic attacks[1]) remains unclear.
- Considering the high computational cost and training budget associated with diffusion models tailored to a single image classifier, CausalDiff may lack practicality. Strategies to reduce these costs should be considered. For instance, could a smaller diffusion model be trained to replace the two different diffusion models (unconditional and conditional) through model distillation?

[1] Ghiasi, Amin, Ali Shafahi, and Tom Goldstein. "BREAKING CERTIFIED DEFENSES: SEMANTIC ADVERSARIAL EXAMPLES WITH SPOOFED ROBUSTNESS CERTIFICATES." *International Conference on Learning Representations*.

---

> ### Author Rebuttal · Authors · 2024-08-07
>
> **We appreciate the time and effort of the reviewer. In response to the issues raised in the review, we offer the following replies:**
>
> **For Weakness 1**:
>
> - **Similarities**:
> Both CausalAdv [1] and DICE [2], like our CausalDiff, model the generative mechanisms of clean data, with causal relationships defined as $S \rightarrow X \leftarrow Z$, and $S \rightarrow Y$.
>
> - **Differences**:
> The defensive strategies vary because they require adversarial examples generated by a specific attack during training, while our CausalDiff does not. Specifically,
>   - CausalAdv [1]: The causal graph explicitly models adversarial perturbations $E$ thus adversarial examples generated by a specific attack are required during training to identify variable $E$.
>   - DICE [2]: In the Pipeline of domain-attack invariant causal learning, DICE is trained on adversarial examples to eliminate spurious correlations between confounders $V$ and labels $Y$ that arise from specific attack behaviors used during training.
>
> **For Weakness 2**: Yes, thank you for your suggestion. We will revise this paragraph to illustrate better the key challenges solved by the proposed method.
>
> **For Question 1**:
> - **What the label-causative feature learned**:
>   - **Visualization of cases**: In Figure 1 (right), we separately visualize the images conditioned on $S$ (with $Z$ masked) and on $Z$ (with $S$ masked) using our CausalDiff. Interestingly, $S$ captures the general semantic concept of a horse, even from just the image of the head, while $Z$ contains the style like skin color. More cases found in Appendix C.3.
>   - **Visualization of feature space**: As shown in Figure 4, the label-causative feature $S$ aligns with the human semantic understanding of categories. Semantically similar categories, such as cats and dogs, are proximate in the $S$ space.
>
> - **Why CausalDiff works**:
>   - Inspired by the robust reasoning abilities of humans, CausalDiff focuses on the semantic factors regardless of any type of adversarial perturbations. If attackers wish to succeed, they must impact the image's semantics, which is inherently more challenging and requires a larger $\epsilon$-budget. Consequently, our method exhibits enhanced robustness.
>
> - **Attack Types**: We consider CausalDiff to be highly robust against various adversarial attacks. It may face challenges with unseen corruptions such as rotation and translation while it actually extends beyond the scope of our paper.
>
> **For Question 2**: The two opposite optimization directions $(1-\lambda) I(X;S,Z)$ in Eq.(5) ensure that the mutual information $I(X;S,Z)$ is **optimized to an appropriate level at the information bottleneck**, allowing $S$ and $Z$ to neither be independent of $X$ nor overly sensitive, thus making $S$ and $Z$ appropriate semantic abstractions of $X$.
>
> - The **positive term** $I(X;S,Z)$ aims to encourage $S$ and $Z$ to adequately represent the information in $X$. It is derived from the goal of maximizing the mutual information between observed data and latent variables in Eq.(3).
> - The **negative term** $\lambda I(X;S,Z)$ aims to avoid exactly matching all details of $X$ and instead encourages $S$ and $Z$ to learn condensed, abstract semantics of $X$. (sorry for the typo in 196, We intended to express "To avoid S and Z containing too many unimportant details of X.")
>
> Regarding the causal modeling, any additional information in $X$ beyond what is captured by $S$ and $Z$ would typically be modeled by **exogenous variables** in a causal model, which is generally not explicitly represented.
>
> **For Question 3**: We evaluated the robustness of our model against three white-box attacks (APGD-ce, APGD-t, FAB-t) as $\lambda$ varied. As $\lambda$ increases for insensitive to pixel perturbation in $X$, model robustness is initially improved. However, when $\lambda$ reaches 0.5, it begins to degrade the feature information in latent factors $s$ and $z$, resulting in decreased robustness.
>
> |$\lambda$|5e-3|1e-2|5e-2|1e-1|5e-1|
> |:-:|:-:|:-:|:-:|:-:|:-:|
> |Robustness(%)|82.81|85.16|85.74|86.52|84.38|
>
> **For Question 4**: Thank you for pointing these out. We will fix these typos immediately.
>
> **For Limitation 1**: Facing unbound attacks [4], all models exhibit significant vulnerabilities, but our model shows a slight advantage (+2.7%) compared to adversarial training.
>
> |Method|Clean Accuracy(%)|Robust Accuracy(%)|
> |-|:-:|:-:|
> |VGG 16|93.6|5.7|
> |Adversarial Training|87.3|8.4|
> |CausalDiff|90.23|11.1|
>
> Typically, unbound attacks may not be prevalent in robustness evaluations because once the perturbation magnitude is large, it becomes easily detectable by defenders. This contradicts the basic assumptions of adversarial examples, which are intended to be subtle and undetectable.
>
> **For Limitation 2**: We appreciate your perspective. Since the primary computational overhead of CausalDiff occurs during causal factor inference by estimating likelihood on 10 timesteps, we implemented a **Distilled CausalDiff** by modifying the last layer of the diffusion process to predict noise for 10 timesteps in a single operation, as illustrated in Figure 1 of the one-page PDF.
>
> The Distilled CausalDiff requires only about 13% of the original inference time while maintaining 95% of the performance of the original CausalDiff, **achieving 82.55% robustness—still more robust than state-of-the-art methods**. For a more details please refer to the global comment.
>
> **We appreciate your efforts and are open to further discussion if you have any additional concerns.**
>
> [1] CausalAdv: Adversarial Robustness through the Lens of Causality, ICLR 2022
>
> [2] DICE: Domain-attack Invariant Causal Learning for Improved Data Privacy Protection and Adversarial Robustness, KDD 2022
>
> [3] beta-VAE: Learning Basic Visual Concepts with a Constrained Variational Framework, ICLR 2017
>
> [4] Semantic Adversarial Examples, CVPR 2018

---

> > ### Comment · Reviewer_pU16 · 2024-08-08
> >
> > Thanks for the detailed response by the authors. My concern has been addressed. I will raise my score to weak accept.

---

> > > ### Author Response · Authors · 2024-08-08
> > >
> > > Thank you for raising the score, and we appreciate your recognition of our work.

---

### Official Review · Reviewer_dVc3 · 2024-07-09

**Soundness:** 3
**Presentation:** 3
**Contribution:** 3
**Rating:** 6
**Confidence:** 3

**Summary:**

This work aims to promote the trustworthiness of DNNs through purification. To this end, the authors propose a novel causality view to perform a disentanglement approach using diffusion models. Some experiments are conducted to verify the effectiveness of the proposed method.

**Strengths:**

+ This work takes a good step towards practical adversarial defense. For instance, adversarial attack behaviors are unpredictable, leading to the requirement for robustness against unseen attacks. In this context, the authors highlight the challenge and propose a novel approach to address the challenge.

+ The experiment designed on the toy dataset is interesting and provides clear points to depict the motivation of this work. Namely, disentanglement is a promising direction to promote the robustness of DNNs. Moreover, this aligns well with Eq. (3), especially the third term on the right.

+ The experimental results are decent, where the proposed method achieves exciting robustness under various settings. Moreover, the proposed method is evaluated using a SOTA adversarial attack method, i.e., AutoAttack. This makes the results convincing.

**Weaknesses:**

- Adaptive attacks are lacking in this work, which significantly weakens its contribution and convincingness. Specifically, in the context of the considered white-box attack, what if the adversary generates adversarial examples using the following objective function?
$\max_{\delta} \ell_{ce}(x+\delta, y) + \log p(x+\delta) + \log p(x+\delta|s^+,z)$ with $s^+ = \arg \min \log p(y|s^+)$

- It is hard to accept the conclusion shown in Figure 2. Specifically, DNNs would exhibit vulnerability when increasing the $\epsilon$-budget. However, Figure 2 shows that the proposed method always makes DNNs robust. It is unclear what would happen if we further improve the budget. The results would be solid evidence for the false robustness if it is not intuitive.

- Adversarial training with specific adversarial examples endows DNNs with robustness against these adversarial examples. This is intuitive. However, the current version of this work lacks a detailed and explicit explanation for the intuition or mechanism of why the proposed method can endow DNNs with robustness against various types of adversarial attacks. Relax; this is just a suggestion to make the work more solid.

- It is known that diffusion models are sensitive to the inference step related to the inference time. Thus, the proposed method would introduce more cost in time. However, the authors seem to overlook the cost.

Minor:
The second and third paragraphs of the Introduction lack references. For instance, the authors should provide corresponding works discussing certified defense, adversarial training, purification methods, and the causality and disentangle view.

- The authors claim that CausalDiff can be viewed as semantic-level purification. This shows a relation to a previous work [1]. Specifically, performing a (non-semantic) subspace exploration would endow DNNs with robustness to distribution shift [1]. Thus, if we can abstract s* and z* using CausalDiff, we can further perform a distribution exploration to promote the robustness for OOD generalization.

[1] Moderately Distributional Exploration for Domain Generalization. Dai et al.

**Questions:**

Please see the weakness.

**Limitations:**

Yes

---

> ### Author Rebuttal · Authors · 2024-08-07
>
> **We appreciate the time and effort of the reviewer. In response to the issues raised in the review, we offer the following replies:**
>
> **Q1**: Adaptive attacks are lacking in this work.
>
> **A1**: **All robustness evaluations in our paper are conducted against adaptive attacks.** This means the attacker has full knowledge of the inference mechanism and model parameters.  The attack objective $\max_{\delta} \ell_{ce}(x+\delta, y)$ is **adaptively applied to each component of CausalDiff, including purification and causal factor inference.**
>
> Specifically, in the case of a gradient-based attack, the attacker generates adversarial examples with the objective $\max_{\delta} \ell_{ce}(x+\delta, y)$ according to the **chain rule**. The gradient for the adaptive attack in each component of CausalDiff is expressed as follows:
>
> $$\frac{\partial \ell_{ce}(x+\delta, y)}{\partial (x+\delta)} = \frac{\partial \ell_{ce}(s^+, y)}{\partial s^+} \cdot \frac{\partial s^+}{\partial x_{\mathrm{purified}}} \cdot \frac{\partial x_{\mathrm{purified}}}{\partial (x+\delta)}$$
>
> Thus, the objective $\max_{\delta} \ell_{ce}(x+\delta, y)$ of the adaptive attack is achieved by optimizing the following three terms:
> - $\frac{\partial \ell_{ce}(s^+, y)}{\partial s^+}$ (corresponds to the sub-objective $\min_{s^+} \log p(y|s^+)$)
> - $\frac{\partial s^+}{\partial x_{\mathrm{purified}}}$ (corresponds to the sub-objective $\max_{\delta} \log p(x_{\mathrm{purified}} | s^+, z)$, where $x_{\mathrm{purified}}$ is generated by $x+\delta$ in consideration of the likelihood maximization)
> - $\frac{\partial x_{\mathrm{purified}}}{\partial (x+\delta)} $(corresponds to the sub-objective $\max_{\delta} \log p(x+\delta)$)
>
> **This aligns well with the aspects the reviewer mentioned that need to be considered simultaneously.** According to your valuable feedback, we will add a more detailed description of the implementation of the adaptive attack in our paper.
>
> **Q2**: It is hard to accept the conclusion shown in Figure 2.
>
> **A2**: In PGD attack, both $\epsilon$-budget and step size $\alpha$ together determine the attack's strength. In our experimental setup, as detailed in Appendix B.3, we use a constant value of $\alpha = 2/255$ for 100 attack steps. Even as $\epsilon$ increases, the actual intensity of the attack struggles to rise further due to the small $\alpha$. However, this does not impact the relative robustness of the models. We will consider replacing this figure with experimental results where $\alpha$ increases proportionally with $\epsilon$.
>
> **Q3**: A detailed and explicit explanation for the intuition or mechanism of why the proposed method can endow DNNs with robustness against various types of adversarial attacks.
>
> **A3**: We appreciate your valuable advice.
>
> - **Intuitions** for why CausalDiff works:
> Inspired by the robust reasoning abilities of humans, our goal is to construct a **robust feature extractor** that captures label-causative features resistant to perturbations and uses these features to predict label $Y$. Thus, such robust features should only focus on the semantic factor regardless of any type of adversarial perturbations. **If attackers want to succeed, they need to impact the image's semantics, which is more challenging**, because it requires a larger $\epsilon$-budget. Consequently, our method achieves better robustness.
>
> - **Evidence** for Why CausalDiff works: Results show that CausalDiff smartly finds the robust feature $S$ for robust prediction. Specifically,
>   - as depicted in the case of a horse shown in Figure 1 (right), we visualized images conditioned on $S$ (with $Z$ masked) and on $Z$ (with $S$ masked) using our CausalDiff. Surprisingly, **$S$ captures the general semantic concept of a horse for this case**, even from just the head of the horse, while $Z$ contains details like the horse’s skin color.
>   - as demonstrated in Figure 4, we visualize the feature space. We found that **$S$ aligns with human semantic understanding of categories**, that is semantically similar categories (e.g., cat and dog) are also close in the $S$ space.
>
> **Q4**: The proposed method would introduce more cost in time.
>
> **A4**: Unlike the image generation task, which requires traversing all timesteps (e.g., 1000 steps) to generate an image from Gaussian noise, our CausalDiff method only needs to sample $N_t$ timesteps to estimate the likelihood via ELBO for inference. Empirically, we have found that $N_t = 10$ is sufficient.
>
> Regarding a **detailed discussion of inference efficiency, please refer to the global response.**
>
>
> **Q5**: The second and third paragraphs of the Introduction lack references.
>
> **A5**: Thank you for your reminder; we will add the relevant references immediately.
>
>
> **Q6**: If we can abstract s* and z* using CausalDiff, we can further perform a distribution exploration to promote the robustness for OOD generalization.
>
> **A6**: We appreciate your valuable advice. We also believe that CausalDiff should naturally address out-of-distribution (OOD) issues if it is trained with a huge amount of data across various domains, as it eliminates the spurious correlation between $z^*$ and label $y$.
>
> As shown in Table 3, we have tested the robustness against fog corruption for traffic sign recognition and observed the potential for improving OOD robustness. According to our experimental observations (e.g., visualization of feature space in Figure 4 and case study in Figure 8), $s^*$ learns semantics aligned with human perception, which should be robust to distribution shifts. In the future, we would like to extend our model on more data and aim to improve robustness to distribution and adversarial attacks simultaneously.
>
> [1] Robust Classification via a Single Diffusion Model, ICML 2024
>
> [2] Diffusion Models for Adversarial Purification, ICML 2022
>
> We appreciate your efforts and are open to further discussion if you have any additional concerns.

---

> > ### Comment · Reviewer_dVc3 · 2024-08-08
> > **Official Comment**
> >
> > I appreciate the authors' detailed responses, which address my concerns. Thus, I raise the score to "weak accept".

---

> > > ### Author Response · Authors · 2024-08-08
> > >
> > > Thank you for raising the score, and we are grateful for your recognition of our work.

---

### Official Review · Reviewer_DuQ4 · 2024-07-11

**Soundness:** 4
**Presentation:** 3
**Contribution:** 4
**Rating:** 8
**Confidence:** 4

**Summary:**

This paper proposed a novel causal diffusion framework based on causal inference to defend against unseen attacks. The causal information bottleneck is interesting to disentangle the target-causative and target-non-causative factors, and then target-causative factors are used for adversarial defense.

**Strengths:**

The originality of this paper is good, since the causal diffusion framework and causal information bottleneck are innovative and effective for adversarial defense. The quality and clarity of this paper are also good, where the logic is clear for understanding. The significance of this paper is obvious, because unseen attacks are necessary to address.

**Weaknesses:**

1. Some technical details need to be explained. For example, what is the actual meaning of the constraint in Eq. (4)? And why are the positions of z_s and s_b defined like in Eq. (2)?

2. Numerically, the authors could consider comparing their method with more baselines. There are some studies on adversarial defense, even without using diffusion models.

**Questions:**

1. Some technical details need to be explained. For example, what is the actual meaning of the constraint in Eq. (4)? And why are the positions of z_s and s_b defined like in Eq. (2)?

2. Numerically, the authors could consider comparing their method with more baselines. There are some studies on adversarial defense, even without using diffusion models.

**Limitations:**

1. What is the technical drawback of the proposed method? E.g., defense efficiency
2. Does this proposed method work for data other than images?

---

> ### Author Rebuttal · Authors · 2024-08-07
>
> **We appreciate the time and effort of the reviewer. In response to the issues raised in the review, we offer the following replies:**
>
> **Q1**: Some technical details need to be explained. For example, what is the actual meaning of the constraint in Eq. (4)? And why are the positions of z_s and s_b defined like in Eq. (2)?
>
> **A1**: The latent factors should **capture the semantic factors** in $x$, **rather than exactly matching** all the details of $x$ (even adversarial perturbation). Therefore, we constrain the mutual information $I(X; S, Z)$ by $I_c$ in Eq. (4) to limit the sensitivity of $s$ and $z$ to pixel perturbations in $x$.
>
> The distinct roles of $z_s$ and $s_b$ dictate their different positions in the control mechanism. Inspired by **class-conditional generation** [6], the label-causative factor $s$ **controls semantics related to the category**, thus acting as a bias that affects the direction of the latent vector. In contrast, the label-non-causative factor $z$ **governs style and background, and can only scale the latent vector, similar to style control** [7].
>
> Thank you for your feedback, we will add more specific details for explanation in the revision.
>
> **Q2**: Numerically, the authors could consider comparing their method with more baselines. There are some studies on adversarial defense, even without using diffusion models.
>
> **A2**: Indeed, most of the baselines in our experiments involve diffusion models for adversarial defense since they achieve state-of-the-art robustness in the domains of adversarial defense due to the capability of diffusion techniques. Comparing our method with these models provides valuable observations and insights.
>
> We also compare our approach with discriminator-based adversarial training (AT series), causality-based methods (CausalAdv, DICE, CAMA), and geometry-based methods (DOA) in Tables 2 and 3.
>
> According to your valuable feedback, we add the following **non-diffusion defense baselines** with the same setting as Table 2:
>
> |       Method      | Architecture         | Clean acc | $\ell_\infty$ | $\ell_2$ | StAdv | Avg  |
> |-----------------|----------------------|:--------:|:-------------:|:--------:|:-----:|:----:|
> | DecoupledKL [2]   | WideResNet-28-10     | 93.16     | 67.97         | 69.53    | 91.21 | 76.24|
> | RobustArchitecture (RA) [3] | RaWideResNet-70-16 | 93.55 | 71.48     | 68.36    | 91.60 | 77.15|
> | MeanSparse [4]    | RaWideResNet-70-16   | 93.36     | 72.27         | 69.92    | 91.80 | 78.00|
> | AT-scaling [5]    | WideResNet-94-16     | 93.95     | 74.41         | 69.53    | 92.38 | 78.77|
> | **Our CausalDiff**    | Diffusion            | 90.23     | 83.01         | 86.33    | 89.84 | **86.39**|
>
>
> **Q3**: What is the technical drawback of the proposed method? E.g., defense efficiency
>
> **A3**: As discussed in the limitations section of Appendix D, handling unseen attacks does indeed incur certain costs. Please refer to the global response regarding a detailed discussion of computational complexity and training costs.
>
> **Q4**: Does this proposed method work for data other than images?
>
> **A4**: This is an excellent idea! We believe that causal models may be applicable to different types of data while diffusion may not. For instance, transformer architecture might be more suitable for text. For text classification tasks, the label-causative factor **$S$ may encompass the semantics of intent**, while the label-non-causative factor **$Z$ could relate to language style or even language type** (which should not influence the task). Similar to CausalDiff, a causal model could be learned from observed data to enhance the adversarial robustness of text classification tasks.
>
> Thank you for the inspiration; this is certainly worth further exploration. Besides, it may demonstrate the generality of our proposed framework in defending against adversarial attacks.
>
> **We appreciate your efforts and are open to further discussion if you have any additional concerns.**
>
> [1] MixedNUTS: Training-Free Accuracy-Robustness Balance via Nonlinearly Mixed Classifiers, Arxiv 2024
>
> [2] Decoupled Kullback-Leibler Divergence Loss, Arxiv 2023
>
> [3] Robust Principles: Architectural Design Principles for Adversarially Robust CNNs, BMCV 2023
>
> [4] MeanSparse: Post-Training Robustness Enhancement Through Mean-Centered Feature Sparsification, Arxiv 2024
>
> [5] Adversarial Robustness Limits via Scaling-Law and Human-Alignment Studies, ICML 2024
>
> [6] Classifier-Free Diffusion Guidance, NeurIPS 2021
>
> [7] Diffusion Autoencoders: Toward a Meaningful and Decodable Representation, CVPR 2022
>
> [8] Robust Classification via a Single Diffusion Model, ICML 2024
>
> [9] Diffusion Models for Adversarial Purification, ICML 2022

---

> > ### Comment · Reviewer_DuQ4 · 2024-08-12
> >
> > Thanks for the authors' efforts in the detailed response. My concern has been addressed. I will raise my score to strong accept.

---

> > > ### Author Response · Authors · 2024-08-13
> > >
> > > Thank you for raising the score, and we appreciate your recognition of our work.

---

### Official Review · Reviewer_t8de · 2024-07-12

**Soundness:** 3
**Presentation:** 3
**Contribution:** 3
**Rating:** 5
**Confidence:** 3

**Summary:**

The authors propose CausalDiff, a causality-inspired disentanglement approach using diffusion models for adversarial defense, which outperforms state-of-the-art methods on various unseen attacks across multiple datasets.

**Strengths:**

Novel approach combining causal modeling and diffusion models for adversarial defense

Strong performance against unseen attacks on multiple datasets

Thorough pilot study on toy data to validate the approach

Clear theoretical foundation with the proposed Causal Information Bottleneck objective

Practical adaptation of diffusion models for conditional generation

**Weaknesses:**

Limited discussion on computational complexity and training time

Lack of comparison with other causal approaches for adversarial robustness

No analysis of the method's performance on more complex datasets (e.g., ImageNet)

Limited exploration of the interpretability of the learned causal factors

Absence of ablation studies to isolate the impact of individual components

**Questions:**

Please see the weaknesses section.

**Limitations:**

The authors adequately addressed the limitations

---

> ### Author Rebuttal · Authors · 2024-08-07
>
> **We appreciate the time and effort of the reviewer. In response to the issues raised in the review, we offer the following replies:**
>
> **Q1**: Limited discussion on computational complexity and training time.
>
> **A1**:  We provide a detailed speed test in Appendix C.4, evaluating the inference time of our CausalDiff and baselines. We will include a discussion on efficiency in the methods section and link to the detailed analysis of efficiency in the appendix. For more details, please refer to the global response regarding the discussion of computational complexity and training costs.
>
> **Q2**: Lack of comparison with other causal approaches for adversarial robustness.
>
> **A2**: We have compared our work with representative and well-recognized studies in the field of causal method for adversarial robustness, including the pioneering work and highly cited articles. As mentioned in lines 40 to 44 and in the related work section, we discuss the differences between other causal approaches and our CausalDiff. Specifically:
>
> - **From a modeling perspective**, other causal approaches handling adversarial robustness like CausalAdv [1], Deep CAMA [2], and DICE [3] model adversarial attack behaviors, aiming to **identify adversarial factors** (e.g., a manipulation variable) through causal mechanisms. This requires learning targeted at specific attack types during training, which limits their robustness across unseen attacks. In contrast, **our CausalDiff models the generative mechanisms** of the genuine data itself in order to enhance adversarial robustness against various unseen attacks.
>
> - **Experimentally**, we compared the robustness of relevant causal approaches, including CausalAdv [1], Deep CAMA [2], and DICE [3], against different adversarial attack methods in Table 2, demonstrating the superiority of our CausalDiff.
>
>
> We hope our response has addressed your concerns. If you think other essential related works should be included, we will add them in the next version.
>
>
> **Q3**: No analysis of the method's performance on more complex datasets (e.g., ImageNet).
>
> **A3**: Due to resource limitations, we have not conducted tests on very large datasets. However, we selected three commonly used datasets for evaluation against various types of attacks and also the robustness against corruption (i.e., fog) in traffic sign recognition. We appreciate your suggestion. We are trying our best to test the effectiveness of CausalDiff on the ImageNet dataset.
>
> **Q4**: Limited exploration of the interpretability of the learned causal factors.
>
> **A4**: We have conducted the following analyses, confirming that S captures the label-causative factor while Z learns the label-non-causative factor:
>
> - **Visualization of distribution of S and Z**: As demonstrated in Figure 4, we leverage T-SNE to display the distributions learned by S and Z. We found that **S aligns with human semantic understanding of categories; semantically similar categories (e.g., cat and dog) are also close in the S space**. In contrast, Z shows a more blurred distinction between categories, which corresponds with our objective to isolate the label-non-causative factors into Z.
>
> - **Visualization of cases**: To analyze what $S$ and $Z$ have learned, we visualized images conditioned on $S$ (with $Z$ masked) and on $Z$ (with $S$ masked) using our trained CausalDiff. As depicted in Figure 1 (right), surprisingly, **$S$ captures the general semantic concept of a horse, even from just the head of the horse**, while $Z$ contains details like the horse’s skin color. For additional cases, see Figure 8 in Appendix C.3.
>
> - **Visualization of $S$ and $Z$ Interpolation from cat to dog**: Additionally, as shown in Figure 2 of the uploaded one-page PDF, we plotted the interpolation of S and Z from a cat to a dog. The evolution process in S shows cat-specific features such as facial characteristics fading, while retaining the animal's body, a commonality between cats and dogs. Conversely, the evolution process in Z includes some unimport information with respect to its category. This evolutionary process aligns with human cognition as expected.
>
> **Q5**: Absence of ablation studies to isolate the impact of individual components.
>
> **A5**: In the last three rows of Table 2, we present the performance of CausalDiff and its individual components, including removing adversarial purification and replacing causal factor inference via a standard discriminator. The corresponding analysis can be found in Section 5.3. The results indicate that each component independently exhibits commendable robustness and is essential; together, they achieve superior robustness.
>
> **We appreciate your efforts and are open to further discussion if you have any additional concerns.**
>
> [1] CausalAdv: Adversarial Robustness through the Lens of Causality, ICLR 2022
>
> [2] A Causal View on Robustness of Neural Networks, NeurIPS 2020
>
> [3] DICE: Domain-attack Invariant Causal Learning for Improved Data Privacy Protection and Adversarial Robustness, KDD 2022

---

> > ### Comment · Reviewer_t8de · 2024-08-13
> >
> > I appreciate the author's effort to address the concerns.
> > I have read the author's rebuttal, and I will maintain my original score.

---

> > > ### Author Response · Authors · 2024-08-13
> > >
> > > Thank you for your reply. We appreciate your time and recognition of our work.

---

### Author Rebuttal · Authors · 2024-08-07

**We appreciate the time and effort of all the reviewers.**

Regarding the inference efficiency mentioned by the reviewers, we provide a detailed speed test in Appendix C.4, evaluating the inference time of our CausalDiff and baselines. We will include a discussion on efficiency in the methods section and link to the detailed analysis of efficiency in the appendix.

We have implemented a distilled version of CausalDiff (marked as Distilled CausalDiff) by modifying the last layer of the diffusion process to predict noise for 10 timesteps in a single operation (as illustrated in Figure 1 of the uploaded one-page PDF).  The comparisons between our methods (CausalDiff and Distilled CasualDiff) and the representative baselines including Robust Diffusion Classifier (RDC) [1] that has SOTA robustness on unseen attacks, in terms of inference time (for a single sample on an NVIDIA A6000 GPU) and average robustness against unseen attacks (consistent with the settings in Table 2) are as follows:


| Method               | CausalDiff | Distilled CausalDiff | RDC [1] | Distilled RDC [1] | DiffPure [2] | WRN-70-16 |
|----------------------|:------------:|:----------------------:|:---------:|:-------------------:|:--------------:|:-----------:|
| Inference Time (s)   | 4.97       | 0.67                 | 19.37   | 15.88             | 2.22         | 0.01      |
| Avg Robustness (%)   | **86.39**      | 82.55                | 82.38   | 78.85             | 47.20        | 0.00      |


**Discussion**:
- We must acknowledge that addressing unseen attacks incurs certain costs. However, when **considering both computational complexity and robustness, our (Distilled) CausalDiff maintains strong competitiveness**, compared to state-of-the-art methods. For instance, compared to the RDC method, our approach requires only about one-quarter of the inference time to achieve superior robustness.
- The **distilled CausalDiff** requires only about 13% of the original inference time while maintaining 95% of the performance of the original CausalDiff, **achieving 82.55% robustness—still more robust than state-of-the-art methods**.
- Regarding **training costs**, training CausalDiff is comparable to training a standard diffusion model. Modeling a causal structure does not introduce additional training overhead but helps us model the generative mechanisms of genuine data. If CausalDiff is trained with tremendous data (similar to StableDiffusion), we would expect that this classifier will be robust in handling unseen attacks. It does not need to construct adversarial examples of all possible types of attacks, and incorporate them into training, which is costly.

[1] Robust Classification via a Single Diffusion Model, ICML 2024

[2] Diffusion Models for Adversarial Purification, ICML 2022

---

### Decision · Program_Chairs · 2024-09-25

**Decision:**

Accept (poster)

**Comment:**

This paper develops a causal diffusion framework (CausalDiff) to improve the adversarial robustness against unseen attacks. A causal information bottleneck is introduced to disentangle label-causative factors and non-causative ones. The effectiveness of CausalDiff is well illustrated by the pilot study. The experimental results are promising, where CausalDiff achieves state-of-the-art robustness under various settings.

All the reviewers found their concerns were adequately addressed after the rebuttal. I therefore recommend acceptance of this paper. I encourage the authors to revise the paper according to the reviewers' comments.